# Marine Pharmacology in 2016–2017: Marine Compounds with Antibacterial, Antidiabetic, Antifungal, Anti-Inflammatory, Antiprotozoal, Antituberculosis and Antiviral Activities; Affecting the Immune and Nervous Systems, and Other Miscellaneous Mechanisms of Action

**DOI:** 10.3390/md19020049

**Published:** 2021-01-21

**Authors:** Alejandro M. S. Mayer, Aimee J. Guerrero, Abimael D. Rodríguez, Orazio Taglialatela-Scafati, Fumiaki Nakamura, Nobuhiro Fusetani

**Affiliations:** 1Department of Pharmacology, College of Graduate Studies, Midwestern University, 555 31st Street, Downers Grove, IL 60515, USA; aguerrero89@midwestern.edu; 2Molecular Sciences Research Center, University of Puerto Rico, 1390 Ponce de León Avenue, San Juan, PR 00926, USA; abimael.rodriguez1@upr.edu; 3Department of Pharmacy, University of Naples Federico II, Via D. Montesano 49, I-80131 Napoli, Italy; scatagli@unina.it; 4Department of Chemistry and Biochemistry, Graduate School of Advanced Science and Engineering, Waseda University, 3-4-1 Okubo, Shinjuku-ku, Tokyo 169-8555, Japan; what-will_be.x2@akane.waseda.jp; 5Fisheries and Oceans Hakodate, Hakodate 041-8611, Japan; anobu@fish.hokudai.ac.jp

**Keywords:** drug, marine, sea, chemical, natural product, pharmacology, pharmaceutical, review, toxicology, pipeline, preclinical

## Abstract

The review of the 2016–2017 marine pharmacology literature was prepared in a manner similar as the 10 prior reviews of this series. Preclinical marine pharmacology research during 2016–2017 assessed 313 marine compounds with novel pharmacology reported by a growing number of investigators from 54 countries. The peer-reviewed literature reported antibacterial, antifungal, antiprotozoal, antituberculosis, and antiviral activities for 123 marine natural products, 111 marine compounds with antidiabetic and anti-inflammatory activities as well as affecting the immune and nervous system, while in contrast 79 marine compounds displayed miscellaneous mechanisms of action which upon further investigation may contribute to several pharmacological classes. Therefore, in 2016–2017, the preclinical marine natural product pharmacology pipeline generated both novel pharmacology as well as potentially new lead compounds for the growing clinical marine pharmaceutical pipeline, and thus sustained with its contributions the global research for novel and effective therapeutic strategies for multiple disease categories.

## 1. Introduction

The present review aims to consolidate the 2016–2017 preclinical marine pharmacology literature, with a format similar to our previous 10 reviews of this series which cover the period 1998–2015 [1,2,3,4,5,6,7,8,9,10]. All peer-reviewed articles were retrieved from the following databases: MarinLit, PubMed, Chemical Abstracts^®^, ISI Web of Knowledge, and Google Scholar. As in our previous reviews we have decided to limit the review to the bioactivity and/or pharmacology of structurally characterized marine chemicals, which we have classified using a modification of Schmitz’s chemical classification [11] into six major chemical classes, namely, polyketides, terpenes, peptides, alkaloids, shikimates, and sugars. The preclinical antibacterial, antifungal, antiprotozoal, antituberculosis, and antiviral pharmacology of marine chemicals is reported in Table 1, with the structures, shown in Figure 1. Marine compounds that showed immune and nervous systems activities, as well as antidiabetic and anti-inflammatory effects are exhibited in Table 2, with their respective structures consolidated in Figure 2. Finally, marine compounds affecting a variety of cellular and molecular targets are noted in Table 3, and their structures presented in Figure 3.

Several publications during 2016–2017 reported on several marine extracts or structurally uncharacterized compounds, with potentially novel *preclinical* and/or *clinical* pharmacology: In vitro antibacterial and antibiotic potentiating activities of tropical Mauritian marine sponge extracts [12]; a sterol-rich fraction with significant synergistic anti-inflammatory activity isolated from the South Korean soft coral *Dendronephthya gigantea* [13]; two metabolites with anti-methicillin-resistant *Staphylococcus aureus* activity from an Indian marine sponge *Clathria procera* associated actinomycete *S. pharmamarensis* ICN40 with antimicrobial potential “after structure identification and clinical studies” are completed [14]; 49 of 251 bacterial isolates from sediments from several Red Sea harbor and lagoon environments with the potential to produce secondary metabolites with “antimicrobial activity” [15]; antibacterial activity in extracts from the marine bacterium *Salinispora arenicola* from the Gulf of California, Mexico [16]; activation by cyanobacterium *Oscillatoria* sp. lipopolysaccharide of rat microglia and murine B cells with concomitant Toll-like receptor 4 signaling in vitro [17,18]; and lipophilic fractions from the Icelandic marine sponge *Halichondria sitiens* that decrease dendritic cells’ pro-inflammatory cytokine release [19].


marinedrugs-19-00049-t001_Table 1Table 1Marine pharmacology in 2016–2017: marine compounds with antibacterial, antifungal, antituberculosis, antiprotozoal and antiviral activities.Drug ClassCompound/Organism ^a^ChemistryPharmacologic ActivityIC_50_
^b1^MMOA ^b2^Country ^c^ReferencesAntibacterialageloline A (**1**)/bacteriumAlkaloid ^f^*C. trachomatis* inhibition9.5 μMAntioxidant activityDEU[20]Antibacterialblasticidin S analog (**2**)/spongeAlkaloid ^f^*S. aureus* inhibition6.2 μg/mL ^+^*norA* multidrug trasporter inactivationCAN, USA[21]Antibacterialdibromohemibastadin-1(**3**)/spongePeptide ^f^*P. auruginosa* biofilm disruption10 μM *Quorum sensing activityDEU, FRA, GBR[22]Antibacterialecteinamycin (**4**)/bacteriumPolyketide ^d^*C. difficile* inhibition0.059 μM ^+^K^+^ transport dysregulationJPN, USA[23]Antibacterial 5-episinuleptolide (**5**)/soft coralTerpenoid ^e^*A. baumannii* biofilm formation inhibition20 μM *PNAG gene expression inhibitionTWN[24]Antibacterial*E. esculentus* peptides (**6**,**7**)/sea urchinPeptide ^f^Gram-positive and negative inhibition0.1–3.1 μM ^+^Heavy chains bioactiveNOR, SWE[25]Antibacterialgranaticin and granatomycin D (**8**,**9**)/bacteriumPolyketide ^d^*B. subtilis* and MR *S. aureus* inhibition1.6, 6.2 μg/mL ^+^Co-culture enhanced MICUSA[26]Antibacterialkeyicin (**10**)/bacteriumPolyketide ^d^*B. subtilis* and MR *S. aureus* inhibition 2.5–9.9 μM ^+^Fatty acid metabolism modulationUSA[27]Antibacterial microcionamides C and D (**11**, **12**)/spongePeptide ^f^*S. aureus* inhibition6.2 μM ^+^Depolarize cytoplasmic membranesDEU, IDN, IRN[28]Antibacterialmyticalin A5 (**13**)/musselPeptide ^f^Gram-positive and negative inhibition2–8 μM ^+^RNA synthesis inhibitionDEU, ITA[29]Antibacterialplakofuranolactone(**14**)/spongePolyketide ^d^
Quorum quenching inhibition0.1 μM *Specificity to QS systemsITA[30]Antibacterial psammaplin A (**15**)/spongePeptide ^f^*V. vulnificus* in vivo growth inhibition50 μg/mouse **Associated in vitro and in vivo pathology suppressedS. KOR[31]Antibacterialabyssomicin 2 (**16**)/bacteriumPolyketide ^d^*B. thuringiensis* and *M. luteus* inhibition3.6, 7.2 μg/mL ^+^UndeterminedCHN[32]Antibacterialactinomycins D, V, and X2 (**17**–**19**)/bacteriumPeptide ^f^MR *S. aureus, B. subtilis* and *E. coli* inhibition0.08–0.61 μMUndeterminedCHN, EGY, SAU[33,34]Antibacterialaneurinifactin (**20**)/bacteriumLipopeptide ^f^*K. pneumoniae* and *S. aureus* inhibition4, 8 μg/mL ^+^UndeterminedIND[35]Antibacterialaspewentins D and H (**21**, **22**)/fungusTerpenoid ^e^*P. aeruginosa* and *M. luteus* inhibition4 μg/mL ^+^UndeterminedCHN[36]Antibacterial*B. subtilis* furanoterpenoid (**23**)/bacteriumTerpenoid ^e^*V. vulnificus* and *parahaemolyticus* inhibition3.12 μg/mL ^+^UndeterminedIND[37]Antibacterialbacillisporin A (**24**)/fungusPolyketide ^d^*B. subtilis* inhibition0.12 μM ^+^UndeterminedBGD, CHN[38]Antibacterial bacilotetrin A (**25**)/bacteriumPeptide ^f^MR *S. aureus* inhibition8 μg/mL ^+^UndeterminedS. KOR[39]Antibacterialbranimycin B (**26**)/bacteriumPolyketide ^d^*M. luteus* and *C. urealyticum* inhibition1, 8 μg/mL ^+^UndeterminedESP[40]Antibacterialbrocazine G (**27**)/fungusAlkaloid ^f^*S. aureus* inhibition0.25 μg/mL ^+^UndeterminedCHN, DEU, HUN[41]Antibacterialcadiolides K and M (**28**, **29**)/ascidianPolyketide ^d^MR *S. aureus* inhibition1–2 μg/mL ^+^UndeterminedS. KOR[42]Antibacterialcahuitamycin D (**30**)/bacteriumPeptide ^f^*A. baumannii* biofilm inhibition8.4 μMUndeterminedCRI, USA[43]Antibacterialchalcomycin (**31**)/bacteriumPolyketide ^d^*S. aureus* inhibition4 μg/mL ^+^UndeterminedCHN[44,45]Antibacterial chermesins A and B (**32**, **33**)/fungusTerpenoid ^e^*M. luteus* inhibition8 μg/mL ^+^UndeterminedCHN[46]Antibacterialchloro-preussomerins A and B (**34**, **35**)/fungusPolyketide ^d^*S. aureus* inhibition3.2, 6.2 μg/mL ^+^UndeterminedCHN[47]Antibacterialcollismycin C (**36**)/bacteriumAlkaloid ^f^MR *S. aureus* biofilm inhibition10 μg/mL *UndeterminedS. KOR[48]Antibacterialengyodontochones A and B (**37**, **38**)/fungusPolyketide ^d^MR *S. aureus* inhibition0.17, 0.24 μMUndeterminedCHN, DEU[49]Antibacterialhydroanthraquinones (**39**–**43**/fungusPolyketide ^d^*S. aureus* inhibition2–8 μg/mL *UndeterminedCHN, DEU[50]Antibacteriallangcoquinone C (**44**)/spongeTerpenoid ^e^*B. subtilis* inhibition6.2 μM ^+^UndeterminedJPN, MMR, VNM[51]Antibacterialluffariellolide (**45**)/spongeTerpenoid ^e^*S. enterica* inhibition4 μg/mL ^+^UndeterminedS. KOR[52]Antibacterialmanzamine alkaloids (**46**–**48**)/spongeAlkaloid ^f^Gram-positive and negative inhibition2–8 ng/mL ^+^UndeterminedIDN, S. KOR[53]Antibacterialnapyradiomycin A_1_ (**49**)/bacteriumTerpenoid ^e^MR *S. aureus* inhibition0.5–1 μg/mL ^+^UndeterminedESP[54]Antibacterialoxysporizoline (**50**)/fungusAlkaloid ^f^MR *S. aureus* inhibition6.25 μg/mL ^+^UndeterminedS. KOR[55]Antibacterial*P. citrinum* 1-(2,6-dihydroxyphenyl)butan-1-one (**51**)/fungusPolyketide ^d^*S. aureus* inhibition6.95 μM ^+^UndeterminedCHN[56]Antibacterialpenicillstressols (**52**, **53**)/fungusPolyketide ^d^MR *S. aureus* inhibition0.5 μg/mL ^+^UndeterminedCHN[57]Antibacterialpestalone (**54**)/fungusPolyketide ^d^MR *S. aureus* inhibition6.25 μM ^+^UndeterminedCHN[58]Antibacterialpestalotionol (**55**)/fungusPolyketide ^d^*B. subtilis* and *S. aureus* inhibition2, 8 μg/mL ^+^UndeterminedCHN, TWN[59]Antibacterialphomaethers A and C (**56**, **57**)/fungusPolyketide ^d^*E. coli* and *S. aureus* inhibition0.15–1.25 μM ^+^UndeterminedCHN[60]Antibacterial4-methyl-3”-prenylcandidusin A (**58**)/fungusPolyketide ^d^MR *S. aureus* and *V. vulnificus* inhibition3.8, 7.8 μg/mL ^+^UndeterminedCHN[61]Antibacterial*Pseudomonas* sp. rhamnolipid (**59**)/bacteriumLipid ^e^*B. cenocepacia* and *S. aureus* inhibition1.6–3.1 μg/mL ^+^UndeterminedGBR, ITA[62]Antibacterialsmenospongine (**60**)/spongeTerpenoid ^e^*B. cereus* and *S. aureus* inhibition3.1 μM ^+^UndeterminedCHN, USA[63]Antibacterial*S. cheonanensis* phthalate (**61**)/bacteriumPolyketide ^d^*P. vulgaris* inhibition4 μg/mL ^+^UndeterminedIND[64]Antibacterialsporalactam B (**62**)/bacteriumPolyketide ^d^MR *S. aureus* and *E. coli* inhibition0.4–1.8 μM **UndeterminedCAN, PHL[65]Antibacterialtetrocarcin A (**63**)/fungusPolyketide ^d^*B. subtilis* inhibition0.03–0.125 μg/mL ^+^Undetermined CHN[66,67]Antibacterialtricepyridinium (**64**)/spongeAlkaloid ^f^*B. subtilis* and *S. aureus* inhibition0.78–1.56 μg/mL ^+^UndeterminedJPN[68]Antibacterialtrocheliane (**65**)/soft coralTerpenoid ^e^*A. baumannii and S. aureus* inhibition 4–4.2 μM ^+^UndeterminedEGY, IDN, SAU[69]Antibacterialtulongicin (**66**)/spongeAlkaloid ^f^*S. aureus* inhibition1.2 μg/mL ^+^UndeterminedITA, NZL, USA[70]Antibacterialvineomycin A_1_ (**67**)/bacteriumPolyketide ^d^*S. aureus* inhibition4 μg/mL^+^UndeterminedCHN[71]Antifungalamphidinol 3 (**68**)/dinoflagellatePolyketide ^d^Pore formation requires cholesterol or ergosterol2.0 μM *Toroidal pore 2.6-4.0 nMJPN, PHL[72,73]Antifungalavarol (**69**)/spongeTerpenoid ^e^*C. albicans* inhibition6–8 µg/mL ^+^UndeterminedSRB[74]Antifungaldihydromaltophilin (**70**)/bacteriumPolyketide ^d^*C. albicans* inhibition3 μMUndeterminedAUS, MEX[75]Antifungalhippolide j (**71a**, **71b**)/spongeTerpenoid ^e^*C. albicans* inhibition0.1 µg/mL ^+^UndeterminedCHN, GBR[76]Antifungalilicicolin H (**72**)/fungusPolyketide ^d^*C. albicans* inhibition<0.25 µg/mL ^+^UndeterminedDEU, DNK, ESP[77]Antifungal iturin F_1_ and F_2_ (**73**, **74**)/bacteriumPeptide ^f^*A. flavus* and *P. griseofulvum* inhibition3.1 µg/mL ^+^UndeterminedJPN, S. KOR[78]Antifungal*P. meleagrinum* macrolides (**75**, **76**)/fungusPolyketide ^d^*C. albicans* inhibition1–2 μg/mL **UndeterminedJPN[79]Antifungalplakinic acid M (**77**)/spongePolyketide ^d^*C. gattii* inhibition2.4 μM **UndeterminedUSA[80]Antifungalpoecillastroside D (**78**)/spongeTerpenoid ^e^*A. fumigatus* inhibition6 μg/mL **UndeterminedESP, FRA, IRL, OMN, SWE[81]Antifungalrocheicoside A (**79**)/ bacteriumAlkaloid ^f^*C. albicans* inhibition4 μg/mL ^+^UndeterminedTUR[82]Antimalarialdiacarperoxide A (**80**)/spongeTerpenoid ^e^*P. falciparum* D6 and W2 strain inhibition1.9–2.0 μMUndeterminedCHN, USA[83]Antimalarial dudawalamide A and D (**81**, **82**)/cyanobacteriumPeptide ^f^*P. falciparum* W2 strain inhibition3.5 μMUndeterminedJOR, PAN, USA[84]Antimalarialeudistidine A (**83**)/soft coralAlkaloid ^f^*P. falciparum* D6 and W2 strain inhibition1.1–1.4 μMUndeterminedCAN, USA[85]Antimalarialnaseseazine C (**84**)/bacteriumAlkaloid ^f^*P. falciparum* 3D7 inhibition3.5 μMUndeterminedAUS[86]Antimalarial*P. opacum β*-carboline (**85**)/ascidianAlkaloid ^f^*P. falciparum* FcB1inhibition3.8 μMUndeterminedFRA, NZL[87]Antimalarialptilomycalin F(**86**)/spongeAlkaloid ^f^*P. falciparum* 3D7 strain inhibition0.23 μMUndeterminedBEL, FRA, CHE, NLD[88]Antimalarialpustulosaisonitrile-1 (**88**)/nudibranchTerpenoid ^e^*P. falciparum* 3D7 strain inhibition1.08 μMUndeterminedAUS, USA[89]Antileishmanial*A*. *Niger* fatty acids (**89**, **90**)/spongeLipid ^e^*L. infantum* inhibition0.17, 0.34 mg/mLTopIB inhibitionESP, USA[90]Antileishmanialdudawalamide D (**82**)/cyanobacteriumPeptide ^f^*L. donovani* inhibition2.6 μMUndeterminedJOR, PAN, USA[84]Antileishmanial*Gorgonia* sp. sterol (**91**) spongeTerpenoid ^e^*L. infantum* inhibition>10 μM *UndeterminedESP, PAN[91]Antileishmanialircinin-1 and 2 (**92**, **93**)/spongeTerpenoid ^e^*L. donovani* inhibition28–31 μMUndeterminedCHE, DEU, ITA, TUR[92]Antitrypanosomaljanadolide (**94**)/cyanobacteriumPeptide ^f^*T. b. brucei* inhibition0.047 μMUndeterminedJPN[93]Antitrypanosomalmalformin A_1_ (**95**)/fungusPeptide ^f^
*T. congolense inhibition*
0.015 µg/mL UndeterminedJPN, PHL[94]Antitrypanosomalrhodozepinone (**96**)/bacteriumAlkaloid ^f^*T. b. brucei* inhibition16.3 µg/mLUndeterminedDEU, EGY[95]Antituberculosismelophlin A (**97**)/spongeAlkaloid ^f^*M. smegmatis* inhibition0.8 μg/mL ^+^BCG1083 & BCG1321c proteins targetedIDN, JPN[96]Antituberculosismethoxypuupehenol (**98**)/spongeTerpenoid ^e^Dormant *M. tuberculosis* inhibition0.5 μg/mL ^+^Bactericidal activityUSA[97]Antituberculosisgliotoxin (**99**)/fungusAlkaloid ^f^*M. tuberculosis* inhibition0.03 μM ^+^UndeterminedCHN[98]Antituberculosisproximicin B (**100**)/bacteriumPeptide ^f^*M. bovis* Pasteur 1173P2 inhibition6.25 μg/mL ^+^UndeterminedAUS, CHN, EGY, NGA[99]Antituberculosissmenothiazole A (**101**)/spongePeptide ^f^*M. tuberculosis* H_37_Rv inhibition4.1 μg/mL ^+^UndeterminedPOL, USA[100]Antituberculosissporalactam B (**62**)/bacteriumPolyketide ^d^*M. tuberculosis* inhibition0.06 μM **UndeterminedCAN, PHL[65]Antituberculosistalaramide A (**102**)/fungusAlkaloid ^f^Mycobacterial PknG inhibition 55 μMUndeterminedCHN[101]Antituberculosisviomellein (**103**)/fungusPolyketide ^d^Dormant *M. bovis* BCG inhibition1.56 μg/mL ^+^UndeterminedIDN, JPN[102]Antiviralhymenialdisine (**104**)/spongeAlkaloid ^f^HIV-1 inhibition>3.1 μM * Reverse transcriptase inhibitionDEU, SAU, USA[103]Antiviralmetachromin A (**105**)/spongeTerpenoid ^e^HBV inhibition0.8 μMViral promoter inhibitionJPN, NLD[104]Antiviralperidinin (**106**)/coralTerpenoid ^e^HTLV-1 infected T cell inhibition0.7-5.4 μMNF-κB inhibitionJPN[105]Antiviralspiromastilactone D (**107**)/fungusPolyketide ^d^H1N1 influenza A virus inhibition6.0 μMHA-sialic acid receptor binding inhibitionCHN, USA[106]Antiviralxiamycin D (**108**)/bacteriumTerpenoid ^e^PEDV virus inhibition0.93 μMVirion structural proteins inhibitionS. KOR[107]Antiviralzoanthone A (**109**)/sea anemoneTerpenoid ^e^DENV-2 virus inhibition19.6 μMRNA pocket tunnel bindingTWN[108] Antiviral*A. polycladia* aromatic sulfate (**110**)/crinoidPolyketide ^d^HCV NS3 helicase inhibition5 μMUndeterminedJPN[109]Antiviral alotaketal C (**111**)/spongeTerpenoid ^e^HIV expression activation1 μM *UndeterminedCAN[110]Antiviralaspergillipeptide D (**112**)/fungusPeptide ^f^HSV-1 inhibition9.5 μMUndeterminedCHN[111]Antiviralaspergilols H and I (**113**, **114**)/fungusPolyketide ^d^HSV-1 inhibition4.7, 6.2 μMUndeterminedCHN[112]Antiviralasteltoxin E (**115**)/fungusPolyketide ^d^H1N1 and H3N2 influenza virus inhibition3.5, 6.2 μMUndeterminedCHN[113]Antiviral*S. verruca* cyclopentenone (**116**)/soft coralPolyketide ^d^HIV infection inhibition5.8 μMUndeterminedCHN, USA[114]Antiviraleutypellazine E (**117**)/fungusAlkaloid ^f^HIV-1 inhibition3.2 μMUndeterminedCHN, DEU[115]Antiviralω-hydroxyemodin (**118**)/fungusPolyketide ^d^HCV NS3 protease inhibition10.7 μMUndeterminedEGY, SAU[116]Antiviralmalformin C (**119**)/fungusPeptide ^f^HIV infection inhibition1.4 μMUndeterminedCHN[117]Antiviralmanzamine A (**120**)/sponge Alkaloid ^f^HSV-1 inhibition 1 μM ^+^UndeterminedUSA[118]Antiviralperidinin (**106**)/zoanthidTerpenoid ^e^Anti-dengue virus 2 inhibition4.5 μMUndeterminedTWN[119]Antiviralstachybonoid A (**121**)/fungusTerpenoid ^e^Dengue virus prM protein expression inhibition25 μMUndeterminedCHN[120]Antiviralsubergorgols T and U (**122**, **123**)/soft coralTerpenoid ^e^H1N1 influenza A virus inhibition35–37 μMUndeterminedCHN, NLD[121]**^a^ Organism**, *Kingdom Animalia*: ascidian (Phylum Chordata); gorgonian, coral, crinoids, sea anemone, zoanthid (Phylum Cnidaria); sea urchin (Phylum Echinodermata), nudibranch (Phylum Mollusca), sponge (Phylum Porifera); *Kingdom Monera*: bacterium, cyanobacterium (Phylum Cyanobacteria); *Kingdom Fungi*: fungus; *Kingdom Protista*: dinoflagellates; **^b1^ IC_50_**: concentration of a compound required for 50% inhibition in vitro, *: estimated IC_50_, **: in vivo study; ^+^ MIC: minimum inhibitory concentration, **^b2^ MMOA**: molecular mechanism of action; **^c^ Country**: AUS: Australia; BEL: Belgium; BGD: Bangladesh; CAN: Canada; CHE: Switzerland; CHN: China; CRI: Costa Rica; DEU: Germany; DNK: Denmark; EGY: Egypt; ESP: Spain; FRA: France; GBR: United Kingdom; HUN: Hungary; IDN: Indonesia; IND: India; IRL: Ireland; IRN: Iran; ITA: Italy; JOR: Jordan; JPN: Japan; MEX: Mexico; MMR: Myanmar; NGA: Nigeria; NLD: The Netherlands; NOR: Norway; NZL: New Zealand; OMN: Oman; PAN: Panama; PHL: Philippines; POL: Poland; SAU: Saudi Arabia; S. KOR: South Korea; SAU: Saudi Arabia; SRB: Serbia; SWE: Sweden; TWN: Taiwan; TUR: Turkey; VNM: Vietnam; **Chemistry**: **^d^** Polyketide; **^e^** Terpene; **^f^** Nitrogen-containing compound; **Abbreviations**: AHL: acylated homoserine lactones; DENV-2: dengue virus type 2; HA: hemagglutinin; HBV: hepatitis B virus; HCV: hepatitis C virus; HSV: herpes simplex virus; MR: methicillin-resistant; PEDV: porcine epidemic diarrhea virus; PknG: mycobacterial protein kinase G; PNAG: polysaccharide poly-β-(1,6)-*N*-acetylglucosamine; TopIB: topoisomerase IB.


## 2. Marine Compounds with Antibacterial, Antifungal, Antiprotozoal, Antituberculosis and Antiviral Activities

Table 1 presents 2016–2017 preclinical pharmacological research on the antibacterial, antifungal, antiprotozoal, antituberculosis, and antiviral activities of marine natural products (**1**–**123**) shown in Figure 1.


Figure 1Marine pharmacology in 2016–2017: marine compounds with antibacterial, antifungal, antiprotozoal, antituberculosis and antiviral activities.
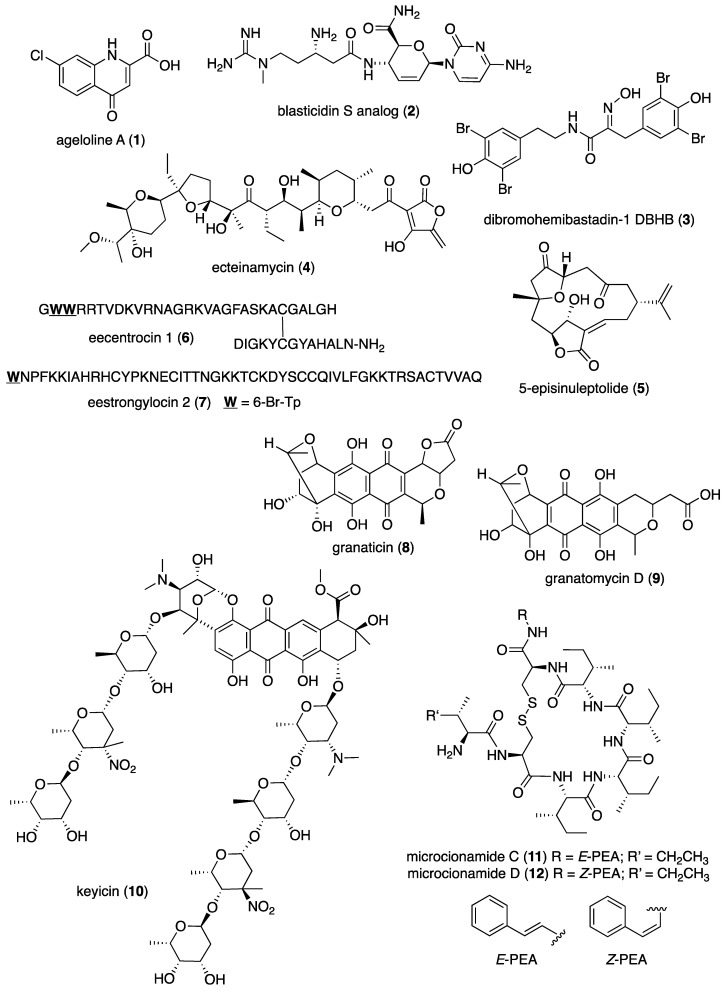

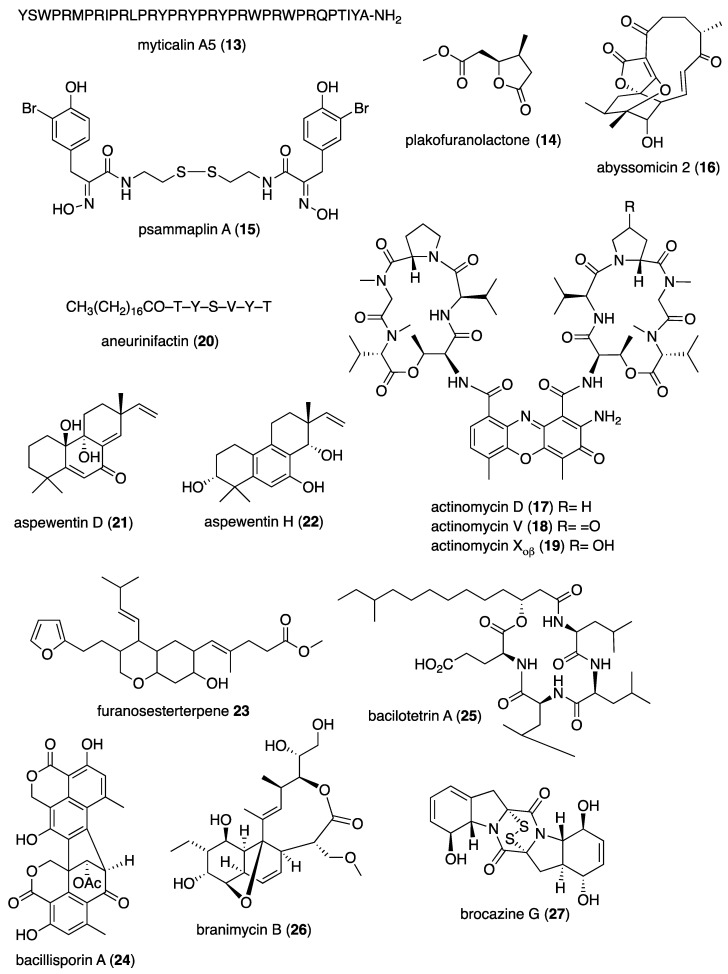

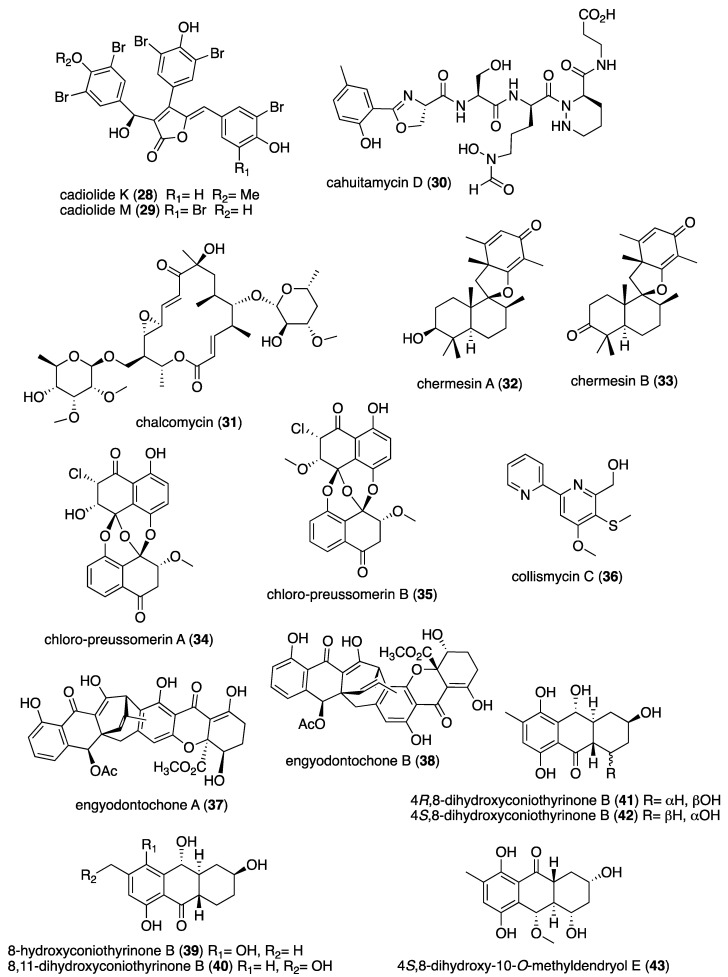

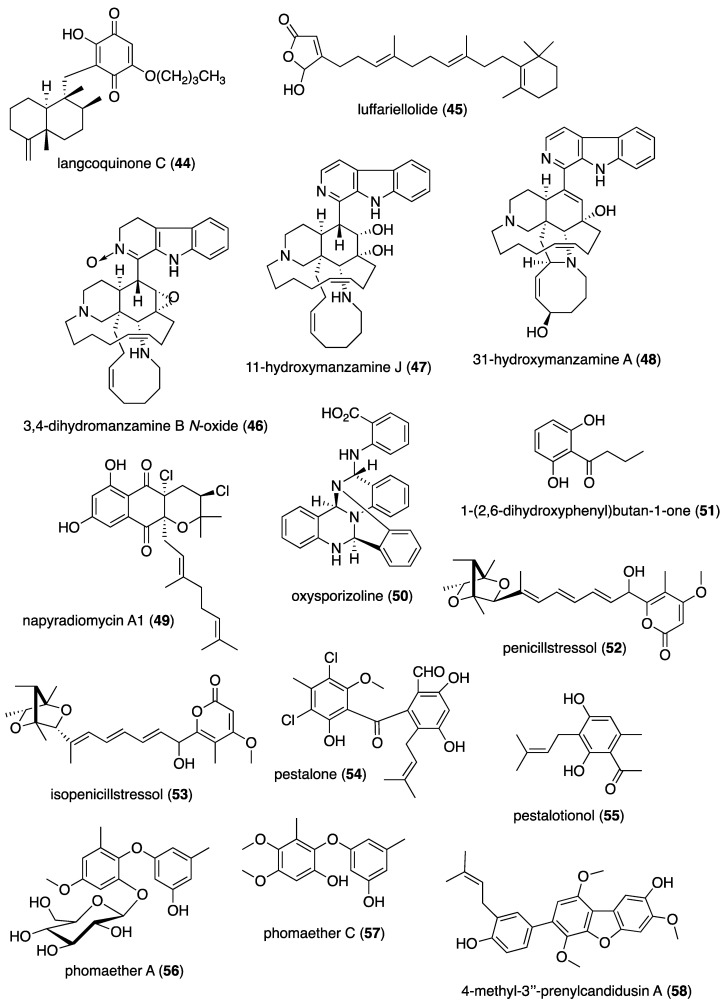

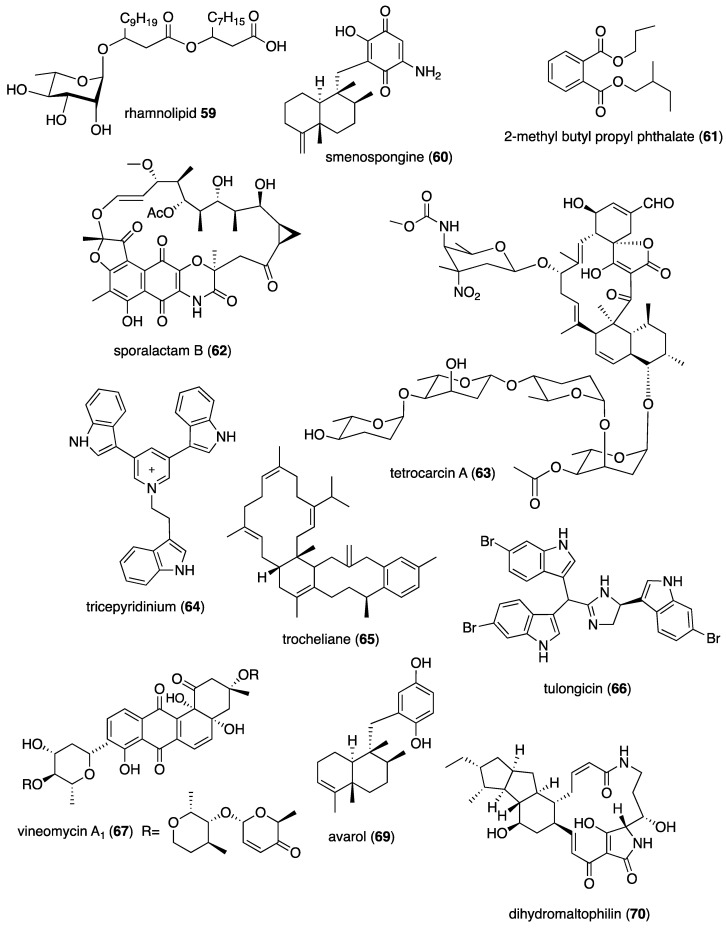

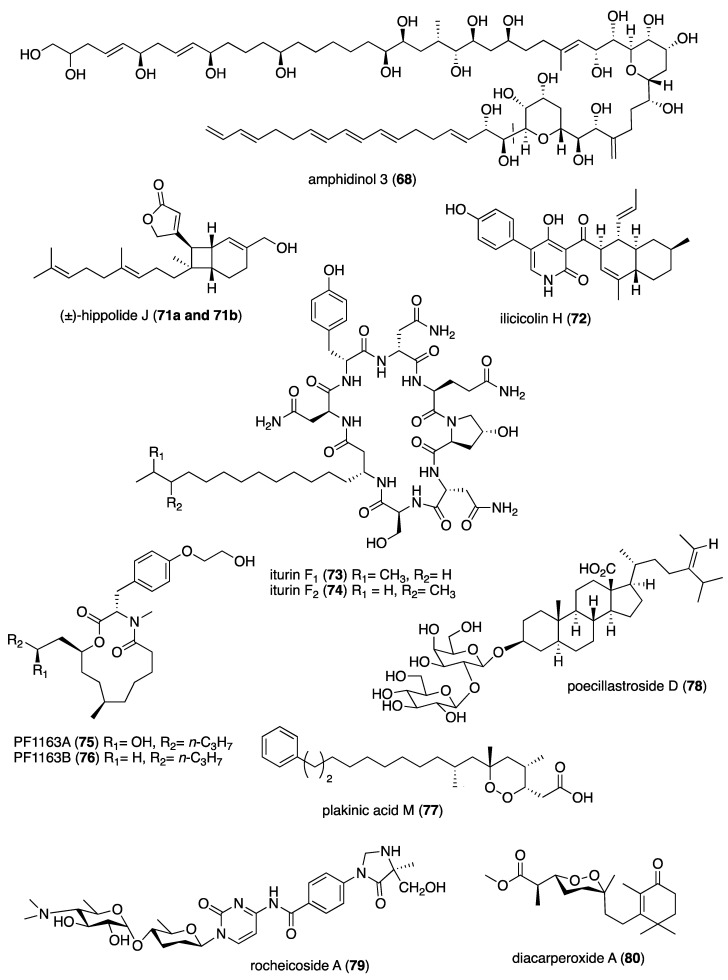

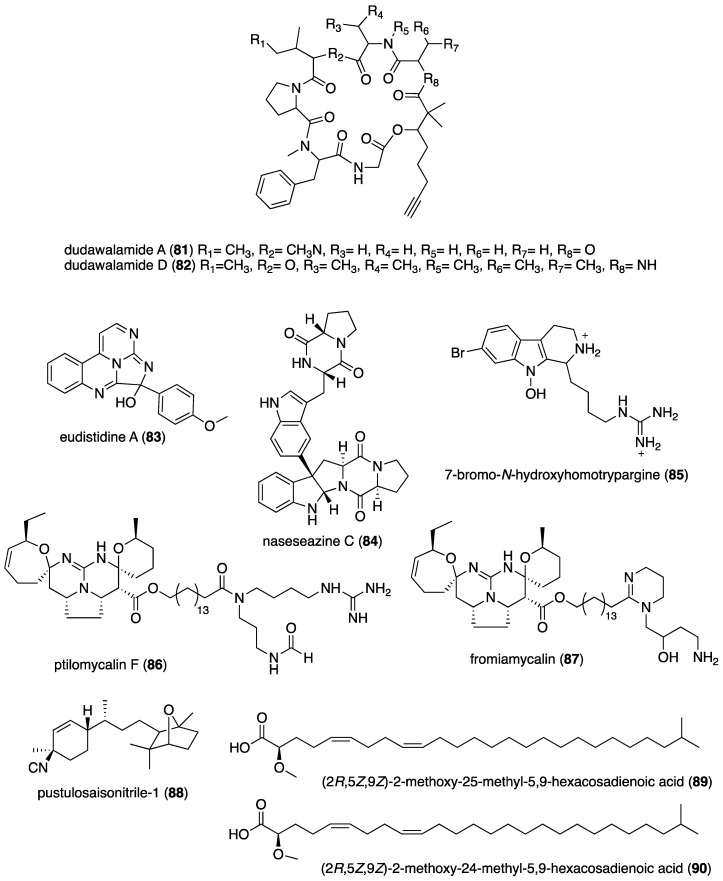

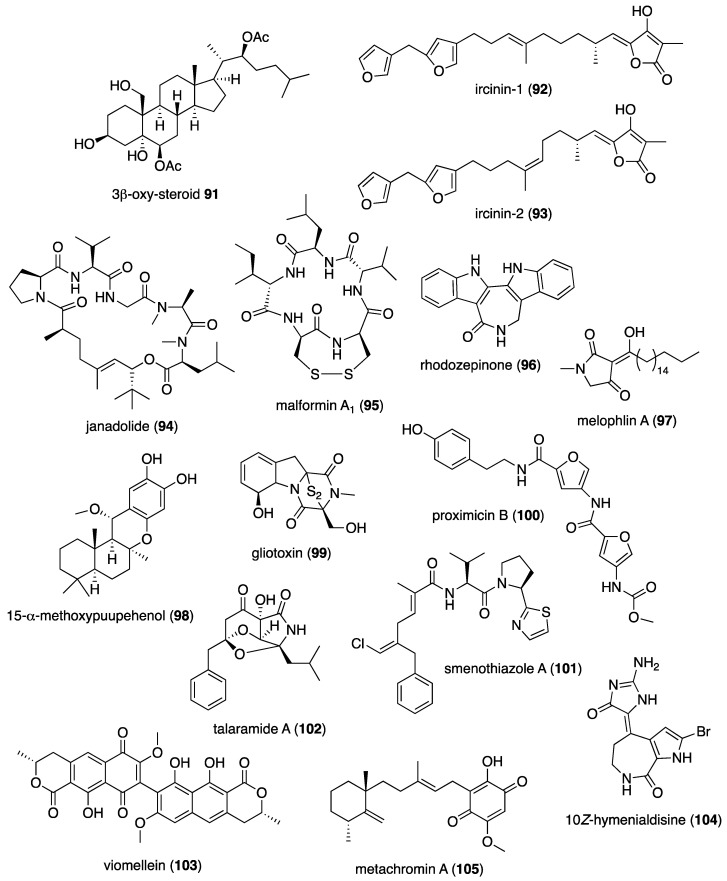

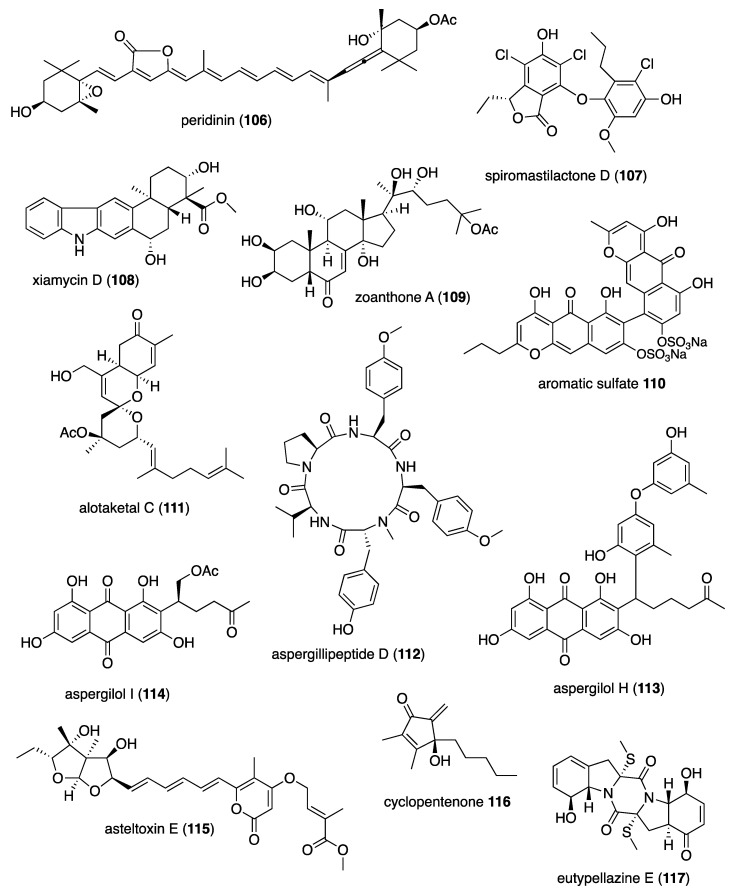

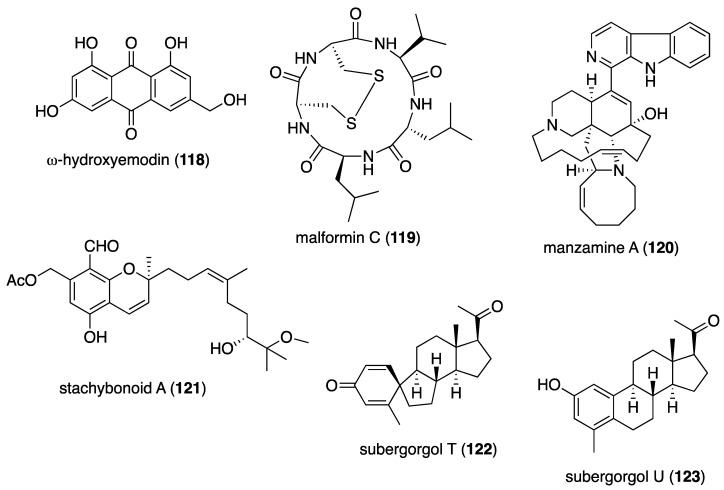



### 2.1. Antibacterial Activity

During 2016–2017, 51 studies reported on *antibacterial* bioactivity in marine natural products (**1**–**67**) isolated from ascidians, bacteria, fungi, mussels, sea urchins, soft corals, and sponges, global research that contributed new preclinical pharmacology that may contribute to the ongoing search of novel therapeutics for multi-drug resistant bacterial infections.

As shown in Table 1, and Figure 1, eleven publications reported on the mode of action of marine-derived antibacterial compounds. Cheng and colleagues reported a new chlorinated quinolone, ageloline A (**1**) isolated from a *Streptomyces* sp. derived from the Mediterranean sponge *Agelas oroides* that inhibited growth of *Chlamydia trachomatis* inclusions resulting from a mechanism that “might be related to its antioxidant potential” [20]. Davison and colleagues discovered a new analog of the peptidyl nucleoside antibiotic blasticidin S (**2**) produced by the actinomycete *Streptomyces griseochromogenes* that demonstrated increased potency against both Gram-positive and Gram-negative bacteria, with the NorA multidrug transporter being a key factor involved in membrane permeability that “facilitates cellular entry of peptidyl nucleosides” [21]. Le Norcy and colleagues described the activity of dibromohemibastadin-1 (**3**), derived from natural bastadins discovered in the marine sponge *Ianthella basta*, and which prevented and disrupted Gram negative bacterial biofilms without toxicity by a mechanism that involved “regulation in the quorum sensing” which is a process considered important for “biofilm formation and organization” [22]. Wyche and colleagues demonstrated in a series of in vitro and in vivo studies that the polyether antibiotic ecteinamycin (**4**) isolated from a marine-derived bacterium *Actinomadura* sp., showed significant activity against the toxigenic strains of *Clostridium difficile* and proposed that the mechanism of action leading to detoxification and cell death likely involved “potassium transport dysregulation” [23]. Tseng and colleagues identified 5-episinuleptolide (**5**) isolated from the soft coral *Sinularia leptoclados* as an inhibitor of biofilm-associated Gram-negative bacterium *Acinetobacter baumannii* infections, shown to be of high incidence in immunocompromised individuals, by a mechanism that correlated with a decreased production of the extracellular polysaccharide poly-β-(1,6)-*N*-acetylglucosamine [24]. Solstand and colleagues described two new antimicrobial peptides eecentrocin 1 and eestrongylocin 2 (**6**, **7**) from the haemocytes of the edible sea urchin *Echinus esculentus* with potent antimicrobial activity against both Gram-positive and Gram-negative bacteria, concluding that “a genomic approach to discover homologues in other echinoderms for the discovery of novel antimicrobial peptides could be a beneficial venture” [25]. Sung and colleagues observed the increased production of the antibiotics granaticin and granatomycin D (**8**, **9**) from marine-derived *Streptomyces* PTY08712 isolated from the tunicate *Styela canopus* when co-cultured with several human bacterial pathogens, concluding that “utilization of co-culture experiments…may enhance metabolite production and further our understanding of … microbial interactions” [26]. Adnani and colleagues reported the isolation of new antibiotic anthracycline keyicin (**10**) from a co-culture of marine bacteria *Rhodococcus* sp. and *Micromonospora* sp. derived from the ascidian *Ecteinascidia turbinata* and the sponge *Chondrilla nucula*, which was found to be selective for Gram-positive bacteria by a unique mechanism that “does not involve nucleic acid damage” [27]. Mokhlesi and colleagues discovered two new peptides microcionamides C and D (**11**, **12**) from the marine sponge *Clathria basilana* which inhibited Gram-positive bacterial growth with a mechanism that suggested “dissipation of the bacterial membrane potential”, which is an electrical gradient across the bacterial cytoplasmatic membrane that is required for both ATP generation and active transport processes [28]. Leoni and colleagues reported that the novel peptide myticalin A5 (**13**) isolated from the mussel *Mytilus galloprovincialis* had antimicrobial activity against both Gram-positive and Gram-negative bacteria by a mode of action that appeared to involve inhibition of RNA synthesis but “remains to be elucidated” [29]. Costantino and colleagues isolated a new γ-lactone plakofuranolactone (**14**) from the extract of the marine sponge *Plakortis* cf. *lita* that inhibited the bacterial LasI/R quorum sensing system, a “mechanism of cell-cell communication and gene regulation in bacteria” [30]. Lee and colleagues studied the activity of psammaplin A (**15**), a marine natural product isolated from sponges, on Gram-negative bacterium *Vibrio vulnificus* infections noting that it suppressed cytotoxicity in vitro and prolonged both the survival and pathology of *V. vulnificus*-infected mice [31].

As shown in Table 1 and Figure 1, 51 marine natural products (**16–67**), some of them novel, were reported to exhibit antibacterial activity with MICs < 10 μg/mL or 10 μM against several Gram-positive and Gram-negative bacterial strains, although their respective mechanisms of action remained undetermined: A polycyclic macrolactone abyssomicin 2 (**16**) isolated from a South China Sea *Streptomyces koyangensis* CSIO 5802 [32]; three bioactive actinomycins (**17–19**) isolated from a marine-derived *Streptomyces* sp. [33,34]; a new lipopeptide aneurinifactin (**20**) produced by the Indian marine *Aneurinibacillus aneurinilyticus* SBP-11 [35]; two new 20-nor-isopimarane diterpenoids aspewenins D and H (**21**, **22**), isolated from a South China Sea marine fungus *Aspergillus wentii* SD-310 [36]; a polyketide furanoterpenoid (**23**) isolated from the Indian brown seaweed-associated heterotrophic bacterium *Bacillus subtilis* MTCC 10403 [37]; a chromone polyketide derivative bacillisporin A (**24**) isolated from a Chinese mangrove-derived marine fungus *Penicillium aculeatum* (No.9EB) [38]; a new cyclic-lipotetrapeptide bacilotetrin A (**25**) isolated from a Korean marine sediment-derived bacterium *Bacillus subtilis* strain 109GGC020 [39]; a new polyketide branimycin B (**26**) isolated from a Cantabrian Sea abyssal (3000 m) actinobacterium *Pseudonocardia carboxydivorans* M-227 [40]; a novel disulfide diketopiperazine alkaloid brocazine G (**27**) was characterized from a mangrove-derived endophytic fungus *Penicillium brocae* MA-231 [41]; two new polyphenyl butenolides cadiolides K and M (**28**, **29**) identified from the Korean marine tunicate *Pseudodistoma antinboja* [42]; a novel peptide cahuitamycin D (**30**) isolated from the Costa Rican marine-sediment derived bacterium *Streptomyces gandocaensis* [43]; a known macrolide chalcomycin (**31**) isolated from a Chinese marine-sediment derived bacterium *Streptomyces* sp. HK-2006-1 [44,45]; two novel spiromeroterpenoids chermesins A and B (**32**, **33**) isolated from the Chinese marine red alga *Pterocladiella tenuis*-derived endophytic fungus *Penicillium chermesinum* EN-480 [46]; two novel chloro-preussomerins A and B (**34**, **35**) isolated from the Chinese mangrove endophytic fungus *Lasiodiplodia theobromae* ZJ-HG1 [47]; a bipyridine collismycin C (**36**) isolated from the Micronesian marine red alga-derived bacterium *Streptomyces* sp. MC025 [48]; two new polyketides engyodontochones A and B (**37**, **38**) isolated from the Croatian marine sponge *Cacospinga scalaris*-derived fungus *Engyodontium album* LF069 [49]; five new polyhydroxylated hydroanthraquinone derivates (**39–43**) isolated from the Chinese marine red alga *Laurencia okamurai*-derived fungus *Talaromyces islandicus* EN-501 [50]; a novel sesquiterpene hydroxyquinone langcoquinone C (**44**) isolated from a Vietnamese marine sponge *Spongia* sp. [51]; the sesterterpene luffariellolide (**45**) isolated from the Philippine marine sponge *Suberea* sp. [52]; three new manzamine alkaloids (**46–48**) isolated from an Indonesian marine sponge *Acanthostrongylophora* sp. [53]; a naphtoquinone terpenoid napyradiomycin A_1_ (**49**) isolated from the Sao Tome and Principe marine ascidian-derived actinomycete *Streptomyces* sp. strain CA-271078 [54]; a new polycyclic quinazoline alkaloid oxysporizoline (**50**) isolated form a Korean marine mudflat-derived fungus *Fusarium oxysporum* [55]; a 1-(2,6-dihydroxyphenyl)butan-1-one (**51**) obtained from the South China Sea mangrove-derived endophytic fungus *Penicillium citrinum* HL-5126 [56]; two novel polyketides penicillstressol and isopenicillstressol (**52**, **53**) isolated from the Chinese marine sediment-derived fungus *Penicillium* sp. BB1122 [57]; a dechlorinated benzophenone polyketide pestalone (**54**) isolated from the South China Sea soft coral-derived fungus *Pestalotiopsis* sp. [58]; a known phenol pestaltionol (**55**) isolated from a Taiwanese marine hydrothermal vent sediment-derived *Penicillium* sp. Y-5-2 [59]; two novel diphenyl ether derivatives phomaethers A and C (**56**, **57**) isolated from a South China Sea gorgonian-derived fungus *Phoma* sp. [60]; a polyketide 4-methyl-3”-prenylcandidusin A (**58**) isolated from the Malaysian marine coral *Galaxea fascicularis*-derived fungus *Aspergillus tritici* SP2-8-1 [61]; a mono-rhamnolipid (**59**) isolated from Ross Sea (Antarctica) sediments-derived *Pseudomonas* sp. BTN1 [62]; the sesquiterpene smenospongine (**60**) isolated from the South China Sea “purple-colored encrusting” marine sponge *Dysidea* sp. [63]; a methyl butyl propyl phthalate (**61**) isolated from Indian mangrove sediments-derived *Streptomyces cheonanensis* VUK-A [64]; a new macrolide sporalactam B (**62**) isolated from Northeastern Pacific Canadian marine sediment-derived *Micromonospora* sp. RJA4480 [65]; a glycosidic spirotetronate polyketide tetrocarcin A (**63**) isolated from a Chinese marine sediment-derived *Micromonospora* sp. 5-297 [66]; a novel pyridinium polyketide tricepyridinium (**64**) was isolated from *Escherichia coli* EPI300 clone pDC113 transfected with metagenomic DNA prepared from the Japanese marine sponge *Discodermia calyx* [68]; a new tetracyclic biscembrane trocheliane (**65**) isolated from the Red Sea soft coral *Sarcophyton trocheliophorum* [69]; a new indole alkaloid tulongicin (**66**) isolated from a Paluan deep-water marine sponge *Topsentia* sp. [70]; and a polyketide vineomycin A_1_ (**67**) isolated from the culture broth of a Taiwan Strait marine sediment-derived actinomycete *Streptomyces* sp. A6H [71].

Furthermore, during 2016–2017, several other marine natural products, some of them novel, reported antimicrobial activity in MICs or IC_50_s ranging from 10 to 50 μg/mL, or 10–50 μM, respectively, and thus because of their lower antibacterial potency were *excluded* from Table 1 and Figure 1: an alkaloid haliclonadiamine from the Okinawan sponge *Halichondria* sp. (MIC = 10 µg/disk) [122]; a sesquiterpene alismol from a Red Sea soft coral *Lobophytum* sp. (MIC = 15 μg/mL) [123]; a new unicellane diterpene eunicellol A from the Arctic soft coral *Gersemia fruticosa* (MIC_90_ = 28–48 μg/mL) [124]; a new cembrane diterpene16-hydroxycembra-1,3,7,11-tetraene (MIC = 25 μg/mL) from the Malaysian soft coral *Sarcophyton* sp. [125]; a new bisindole alkaloid hyrtinadine D (MIC = 16 μg/mL) isolated from the Okinawan marine sponge *Hyrtios* sp. [126]; a briarane diterpenoid dichotelide O (MIC = 2 mg/mL) isolated from the South China Sea gorgonian *Dichotella gemmacea* [127]; two new sesquiterpene aminoquinones langcoquinones A and B isolated from the Vietnamese sponge *Spongia* sp. (MIC = 12.5 μM) [128]; a new lobane diterpenoid prenyl-α-elemenone (MIC = 20 μg/mL) from a Malaysian soft coral *Sinularia* sp. [129]; a new diterpene scaffold darwinolide (MIC = 33.2 μM) from the Antarctic sponge *Dendrilla membranosa* [130]; echinochrome A and spinochrome C (MIC = 22.5 μM) from sea urchins *D. savignyi*, *T. gratilla*, *E.mathaei* and *T. pileolus* from Madagascar [131], the meroterpenoid verruculide B2 (MIC = 32 μg/mL) from the fermentation broth of *Penicillium* sp. SCS-KFD09 isolated from the Chinese marine worm *Sipunculus nudus* [132]; a bioactive sterol (MIC = 20 μg/mL) from a Red Sea soft coral *S. terspilli* [133]; a new norditerpene citrovirin (MIC = 12.4 μg/mL) isolated from a marine algicolous funcus *Trichoderma citrinoviride* [134]; two new bisabolene sesquiterpenes asperchondols A and B (MIC = 25 μM) isolated from the marine sponge *Chondrilla nucula* [135]; long-chain peptaibols peptides (MIC = 25 μg/mL) by a French marine blue mussel-derived strain of *Trichoderma longibrachiatum* [136]; a polyketide antibiotic haliangicin (MIC > 32 μg/mL) and its analogues isolated from the marine myxobacterium *Haliangium ochraceum* SMP-2 [137]; two curvalin macrolides (MIC = 20 μg/mL) isolated from the broth of a cultured marine actinomycete *Pseudonocardia* sp. HS7 [138]; a new halimane-type diterpenoid micromonohalimane B (MIC = 40 μg/mL) isolated from a *Micromonospora* sp. cultivated from the marine ascidian *S. brakenhielmi* [139]; a azaphilonidal derivative penicilazaphilone C (MIC = 15.6 μg/mL) isolated from the marine fungus strain *Penicillium sclerotiorum* M-22 [140]; a new 2′-acetoxy-7-chlorocitreorosein anthraquinone (MIC = 10 μM) from a South China Sea endophytic marine fungus *Penicillium citrinum* HL-5126 [141]; a new antibacterial macrolide borrelidin C (MIC = 6 μM) from a saltern-derived holophilic *Nocardiopsis* sp. [142]; a new antichlamydial dimeric indole derivative (IC_50_ = 46.6–96.4 μM) isolated from the South China Sea sponge-derived actinomycete *Rubrobacter radiotolerans* [143]; the anthraquinone emodin (MIC = 32 μg/mL) isolated from a culture of the endophytic fungus *Eurotium chevalieri* KUFA 0006 [144]; new thiodiketopiperazines eutypellazines P-R (MIC = 16–32 μM) isolated form a deep see fungus *Eutypella* sp. [145]; a new lipophilic cyclic hexapeptide thermoactinoamide A (MIC = 35 μM) isolated from the Icelandic thermophilic bacterium *Thermoactinomyces vulgaris* [146] ; and a new neo-actinomycin A (MIC = 16 μg/mL) isolated from a Chinese marine-derived *Streptomyces* sp. IMB094 [147].

### 2.2. Antifungal Activity

Eleven studies during 2016–2017 reported on the *antifungal* activity of several marine natural products (**68–79**) isolated from marine bacteria, dinoflagellates, fungi and sponges, a slight increase from our last review [10], and previous reviews of this marine pharmacology series.

As shown in Table 1 and Figure 1, two reports investigated an antifungal marine compound with a novel mechanism of action. Espiritu investigated the polydroxy polyene antifungal amphidinol 3 (**68**) isolated from cultures of the marine dinoflagellate *Amphidinium klebsii* [72], observing that membrane integrity loss by direct interaction with membrane lipids showed “an absolute dependence on the presence of sterols”. Furthermore, Iwamoto and colleagues, using atomic force microscopy, determined that amphidinol 3 formed “different types of sterol-aided polymorphic channels in a concentration dependent manner” [73]. 

As shown in Table 1 and Figure 1, ten marine natural products (**69–79**) showed antifungal activity with MICs that were either less than 10 μg/mL, 10 μM, or 10 μg/disk but no mechanism of action studies were reported in the papers: a sesquiterpene avarol (**69**) isolated from the Mediterranean sponge *Dysidea avara* [74]; a polycyclic tetramic macrolactam dihydromaltophilin (**70**) isolated from a marine *Conus miles*-derived *Streptomyces* sp. CMB-CS038 [75]; a pair of enantiomeric sesterterpenoids (±) hippolide J (**71a, 71b**) isolated from the South China Sea sponge *Hippospongia lachne* [76]; a hybrid polyketide ilicicolin H (**72**) isolated from the Danish marine-derived fungus *Stilbella fimetaria* [77] ; new cyclic lipopetides iturin F1 and iturin F2 (**73, 74**) from the Korean saltern-derived marine *Bacillus* sp. KCB14S006 [78]; two macrolide polyketides PF1163A (**75**) and -B (**76**) isolated from a Japanese unidentified marine alga-derived fungus *Penicillium meleagrinum* var. *viridiflavum* [79]; a terpene plakinic acid M (**77**) isolated from the Bahamian sponge association *Plakortis halichondrioides*-*Xestospongia deweerdtae* [80]; a new steroidal saponin poecillastroside D (**78**) isolated from a Mediterranean deep-sea sponge *Poecillastra compressa* [81]; a nucleoside analog rocheicoside A (**79**) isolated from a Turkish marine sediment-derived actinomycete *Streptomyces rochei* 06CM016 [82]. Further characterization of the antifungal pharmacology of these marine-derived natural compounds will require mechanism of action studies. 

In addition, several novel structurally-characterized marine molecules with antifungal MICs or IC_50_s greater than 10 μg/mL, 10 μM, or 10 μg/disk, have been *excluded* from Table 1 and Figure 1 because of their weaker bioactivity: a aminobisabolene (IC_90_ = 17–72 μM) isolated from the tropical Micronesian sponge *Halichondria* sp. [148]; several new bromopyrrole alkaloids (apparent IC_50_ = 20 μM) isolated from the South China sea sponge *Agelas* sp. [149]; and a novel polyketide kalkipyrone B (IC_50_ = 13.4 μM) isolated from American Samoan cyanobacterium *Leptolyngbya* sp. [150]. 

### 2.3. Antiprotozoal and Antituberculosis Activity

As shown in Table 1, and as reported in the 1998–2015 marine pharmacology reviews of this series [1,2,3,4,5,6,7,8,9,10], 22 studies contributed novel findings in 2016–2017 on the *antiprotozoal (antimalarial, antileishmanial and antitrypanosomal)* and *antituberculosis* pharmacology of structurally characterized marine natural products (**80**–**103**). 

Malaria, a global disease caused by protozoan genus *Plasmodium* (*P. falciparum*, *P. ovale*, *P. vivax* and *P. malariae*), currently affects over 2 billion people worldwide. Further contributing to the global antimalarial drugs research, 8 marine molecules (**80**–**88**) isolated from bacteria, molluscs, sponges, soft corals, and ascidians were shown to possess *antimalarial activity* during 2016–2017. As shown in Table 1 and Figure 1, potent (IC_50_ < 2 µM) to moderate (IC_50_ > 2–10 µM) *antimalarial* activity was reported for several marine natural products (**80**–**88**) although the mechanism of action for these compounds remained undertermined at the time of publication: a norterpene cyclic peroxide diacarperoxide A (**80**) isolated from the South China Sea sponge *Diacarnus megaspinorhabdosa* shown to be a potent inhibitor of both *P. falciparum* W2 and D6 clones [83]; cyclic depsipeptides dudawalamides A and D (**81, 82**) from the Papua New Guinean cyanobacterium *Moorea producens* with moderate antimalarial activity against chloroquine-resistant *P. falciparum* strain W2 [84]; the alkaloid eudistidine A (**83**) isolated from Paluan marine ascidian *Eudistoma* sp. that potently inhibited chlorquine-sensitive *P. falciparum* strain D6 and chloroquine-resistant strain W2 [85]; a novel dimeric diketopiperazine naseseazine C (**84**) isolated from an Australian marine-sediment derived *Streptomyces* sp. that exhibited moderate inhibitory activity against chloroquine-sensitive *P. falciparum* strain 3D7 [86]; a new N-hydroxylated 1,2,3,4-tetrahydro-*β*-carboline alkaloid analogue of the known 7-bromohomotrypargine (**85**) isolated from the New Zealand ascidian *Pseudodistoma opacum* with moderate activity against *P. falciparum* chloroquine-resistant strain FcB1 [87]; a new guanidine alkaloid ptilomycalin F (**86**) and known fromiamycalin (**87**) from the Madagascar marine sponge *Monanchora unguiculata* which demonstrated potent activity against *P. falciparum* strain 3D7 [88], and a new isocyanoditerpene pustulosaisonitrile-1 (**88**) from the Australian nudibranch *Phyllidiella pustulosa* which showed potent activity against *P. falciparum* strain 3D7 [89].

As shown in Table 1 and Figure 1, 9 marine compounds (**83**, **89**–**96**) isolated from bacteria, fungi, sponges were reported to possess bioactivity towards the so-called neglected protozoal diseases: Leishmaniasis, caused by the genus *Leishmania (L.*), amebiasis, trichomoniasis, as well as African sleeping sickness (caused by *Trypanosoma (T.) brucei rhodesiense* and *T. brucei gambiense*) and American sleeping sickness or Chagas disease (caused by *T. cruzi*). 

Only one report described two *antitrypanosomal* marine chemicals as well as their mechanisms of action. Carballeira and colleagues examined the mode of action of novel very long-chain α-methoxylated fatty acids (**89**, **90**), isolated from the Caribbean sponge *Asteropus niger,* and demonstrated that they were toxic towards *L. infantum* amastigotes and free living promastigotes by inhibition of *Leishmania* topoisomerase 1B enzyme considered “*an important therapeutic target against L. infantum*” [90]. In addition, seven marine natural products (**82**, **91**–**96**) exhibited *antileishmanial* and *antiprotozoal* activity, although their mechanisms of action remained undetermined: The cyclic depsipeptide dudawalamide D (**82**) from the Papua New Guinean cyanobacterium *Moorea producens* with potent against *L. donovani*. [84]; a new oxysterol (**91**), isolated from a Panamanian octocoral *Gorgonia* sp. that moderately reduced the multiplication of *L. infantum* promastigotes thus suggesting “antileishmanial efficacy against intracellular amastigotes” [91]; two linear furanosesterterpenoids ircinin-1 and -2 (**92**, **93**) from the Turkish sponge *Ircinia oros* with moderate activity against *L. donovani*, suggesting the compounds’ “bifuran terminus… positively influences the in vitro antiprotozoal activity” [92]; a new cyclic polyketide-peptide hybrid janadolide (**94**) from the Japanese marine cyanobaterium *Okeania* sp. which demonstrated very potent activity against *T. brucei brucei,* thus revealing potential for development as “new antitrypanosomal drugs” [93]; the cyclic pentapeptide malformin A1 (**95**) isolated from the Philippine marine seagrass-derived fungus *Aspergillus tubingensis* IFM 63452 highly active towards the parasite *T. congolense* and recommended as ”an antiprotozoal agent” [94]; a novel azepino-diindole alkaloid rhodozepinone (**96**) isolated from a Red Sea marine sponge-derived bacterium *Rhodococcus* sp. UA13 s with moderate activity against *T. brucei brucei* TC221 and perhaps a “promising future contribution to drug discovery” [95]. 

The emergence of drug-resistant *Mycobacterium tuberculosis* has continued to stimulate an ongoing global search for novel therapeutic leads with novel mechanisms of action, and, as shown in Table 1 and Figure 1, during 2016–2017, 8 novel marine natural products (**62**, **97**–**103**), isolated from bacteria, sponges and fungi, generated promising pharmacological activity and thus contributed to the search for novel antituberculosis agents. Arai and colleagues identified the alkaloid melophlin A (**97**) isolated from the Indonesian marine sponge *Melophlus* sp. that demonstrated strong inhibitory activity against dormant *M. smegmatis* by targeting the “BCG1083 protein of putative exopolyphosphatase and the BCG1321c protein of diadenosine 5′,5”-P^1^,P^4^-tetraphosphate phosphorylase” [96]. Rodrigues Felix and colleagues isolated the polyketide 15-α-methoxypuupehenol (**98**) from the marine sponge *Petrosia* sp. that demonstrated potent antibacterial activity against dormant *M. tuberculosis*, highlighting a mode of action in which bacterial killing “is observed only for dormant but not metabolically active bacteria” [97].

As shown in Table 1 and Figure 1, additional six marine natural products (**99**–**103**) exhibited *antituberculosis* activity, although their mechanisms of action remained undetermined: the alkaloid gliotoxin (**99**) derived from a deep-sea fungus *Aspergillus* sp. SCSIO Ind09F01 that strongly inhibited “at very low µM level” *M. tuberculosis* in vitro [98]; the polyketide proximicin B (**100**) isolated from the South China Sea sediments-derived *Verrucosispora* sp. MS100047 which demonstrated “a good anti-BCG activity” [99]; the hybrid peptide/polyketide smenothiazole A (**101**) isolated from a sponge consortium of Puerto Rican marine sponge *Plakortis symbiotica*-*Xestospongia deweerdtae* and identified as a “new lead compound with high activity” against *M. tuberculosis* H_37_Rv [100]; a new macrolide sporalactam B (**62**) isolated from Northeastern Pacific Canadian marine sediment-derived *Micromonospora* sp. RJA4480 reported to demonstrate “selective and potent inhibition of *M. tuberculosis*” [65]; an “unusual” alkaloid talaramide A (**102**) isolated from the mangrove endophytic fungus *Talaromyces* sp. HZ-YX1 that inhibited a mycobacterial protein kinase C required for localization of mycobacterial in macrophage [101], and a naphthoquinone dimer viomellein (**103**) produced by the Indonesian sponge-derived *Aspergillus* sp. that showed potent activity against dormant *M. bovis* BCG [102].

### 2.4. Antiviral Activity

As shown in Table 1 and Figure 1, 18 reports were published during 2016–2017 on the *antiviral* pharmacology of marine natural products (**104**–**123**) against dengue virus, human immunodeficiency virus type-1 (HIV-1), human T-cell leukemia virus type 1 (HTLV-1), human herpes simplex virus (HSV), influenza virus, hepatitis C virus, and porcine epidemic diarrhea virus. 

As shown in Table 1, 6 reports described antiviral marine chemicals and their mechanisms of action. O’Rourke and colleagues communicated that the alkaloid hymenialdisine (**104**), isolated from the Red Sea sponge *Stylissa carteri* inhibited HIV infection and while the retroviral reverse transcriptase was not inhibited, the investigators concluded that it could “serve as starting scaffold(s) for further investigation” [103]. Yamashita and colleagues discovered that the merosesquiterpene metachromin A (**105**) isolated from the marine sponge *Dactylospongia metachromia* significantly inhibited the production of hepatitis B virus (HBV) by affecting the activities of the viral core promoter and reducing the hepatic nuclear factor α protein, a mechanism that may contribute to “ameliorating HBV-related disorders in the liver” [104]. Ishikawa and colleagues reported that the carotenoid peridin (**106**) isolated from the Japanese coral *Isis hippuris* inhibited the proliferation and survival of HTLV-1-infected T-cell lines by a mechanism that involved “suppression of NF-κB and Akt signaling” suggesting the compound was a “promising drug for HTLV-1-associated diseases” [105]. Niu and colleagues determined that the phenolic lactone spiromastilactone D (**107**) isolated from a South Atlantic deep-sea (2869 m) sediment-derived fungus *Spiromastix* sp. MCCC 3A00308 demonstrated “broad anti-influenza spectrum” by a mechanism that targeted viral attachment and entry by affecting “hemagglutinin protein-sialic acid receptor interaction” and viral genome replication by “targeting the viral RNP complex” [106]. Kim and colleagues investigated the indolosesquiterpenoid xiamycin D (**108**) isolated from the Korean saltern-derived halophilic actinomycete *Streptomyces* sp. strain HK18 and observed it displayed potent inhibition of porcine epidemic diarrhea virus by inhibiting genes encoding several essential structural proteins required for PEDV replication, thus demonstrating novel and “promising skeletons against PEDV-related viruses” [107]. Cheng and colleagues isolated a new ecdysone terpenoid zoanthone A (**109**) from a Taiwanese sea anemone *Zoanthus* spp. that demonstrated good activity against dengue virus 2 by a mechanism that inhibited viral replication by blocking the C-terminal RNA-dependent RNA polymerase domain of NS5, the “largest and the most conserved (non-structural) protein” of the virus [108].

An additional 15 marine natural products (**106**, **110**–**123**), listed in Table 1 and shown in Figure 1, demonstrated antiviral activity, but the mechanism of action of these compounds remained undetermined at the time of publication: A new aromatic terpenoid (**110**) isolated form the Japanese marine crinoid *Alloeocomatella polycladia* which showed moderate activity against the hepatitis C virus (HCV) NS3 helicase [109]; a known sesterterpenoid alotaketal C (**111**) isolated from the Canadian marine sponge *Phorbas* sp. shown to activate latent HIV-1 provirus expression [110]; a new cyclic pentapeptide aspergillipeptide D (**112**) isolated from the South China Sea gorgonian *Melitodes squamata*-derived fungus *Aspergillus* sp. SCSIO 41501 with moderate activity against herpes virus simplex type 1 (HSV-1) [111]; new anthraquinones aspergilols H and I (**113**, **114**), isolated from a South China Sea deep sea sediment (2326 m)-derived fungus, *Aspergillus versicolor* SCSIO 41502, with moderate activity against HSV-1 [112]; a new polyketide asteltoxin E (**115**) isolated from a Chinese marine sponge *Callyspongia* sp.-derived fungus *Aspergillus* sp. SCSIO XWS02F40 with moderate activity against influenza virus subtype H1N1 and H3N2 [113]; a novel cyclopentenone derivative (**116**) was isolated from the South China Sea soft coral *Sinularia verruca* which was “moderately protective” against the cytopathic activity of in vitro HIV-1 infection [114]; a new thiodiketopiperazine-type alkaloid eutypellazine E (**117**) isolated form a South Atlantic deep sea (5610 m) sediment-derived fungus *Eutypella* sp. MCCC 3A00281 moderately inhibited in vitro HIV-1 infection [115]; a tricyclic anthraquinone ω-hydroxyemodin (**118**) isolated from the Red Sea brown alga *Padina pavonica*-derived fungus *Fusarium equiseti* moderately inhibited hepatitis C virus NS3/4A serine protease in vitro [116]; the known polyketide malformin C (**119**) isolated from the marine-derived fungus *Aspergillus niger* SCSIO JcswF30 potently inhibited HIV-1 infection in vitro [117]; the known marine β-carboline alkaloid manzamine A (**120**) isolated from the Indo-Pacific sponge *Acanthostrongylophora* sp. potently inhibited HSV-1 replication and release in vitro, observing “that manzamine A had optimal structure features for anti-HSV-1 activity” [118]; the carotenoid peridin (**106**) isolated from the Taiwanese zoanthid *Palythoa mutuki* moderately inhibited all serotypes of dengue virus in vitro [119]; a new meroterpenoid stachybonoid A (**121**) isolated from the Chinese marine crinoid *Himerometra magnipinna*-derived fungus *Stachybotrys chartarum* 952, which evidenced moderate activity against denge virus replication and expression of prM protein [120], and two new pregnane-type steroids subergorgol T and U (**122**, **123**) isolated from a South China Sea gorgonian coral *Subergorgia suberosa* that moderately inhibited influenza virus strain A/WSN/33 (H1N1) in vitro [121].

## 3. Marine Compounds with Antidiabetic and Anti-Inflammatory Activity, and Affecting the Immune and Nervous System

Table 2 presents the 2016–2017 preclinical pharmacology of marine chemicals (**124**–**234**), which demonstrated either antidiabetic or anti-inflammatory activity, as well as affected the immune or nervous system, and whose structures are depicted in Figure 2.


marinedrugs-19-00049-t002_Table 2Table 2Marine pharmacology in 2016–2017: marine compounds with antidiabetic and anti-inflammatory activity; and affecting the immune and nervous system.Drug ClassCompound/Organism ^a^ChemistryPharmacological ActivityIC_50_
^b^MMOA ^c^Country ^d^ReferencesAntidiabeticagelasine G (**124**)/spongeAlkaloid-terpenoid ^f^PTP1B inhibition15 µM *Akt insulin pathway increaseIDN, JAP[151]AntidiabeticBDDE (**125**)/algaShikimate ^h^Decrease glucose levels in vivo10 mg/kg **PTP1B expression inhibitionCHN[152]Antidiabeticdieckol (**126**)/algaShikimate ^h^Decrease in glucose levels1 µg/g **Akt insulin pathway increaseS. KOR[153]Antidiabeticgombasterol E (**127**)/spongeTerpenoid ^f^
Enhanced glucose uptake in vitro 20 µM *AMPK phosphorylation increaseS. KOR[154]Antidiabeticleptolide (**128**)/soft coralTerpenoid ^f^Murine glucose tolerance and insulin sensitivity increased 0.1 mg/kg **PKB phosphorylationESP[155]Antidiabeticnectriacids B and C (**129**, **130**)/fungusPolyketides ^d^α-glucosidase inhibition23.5, 42.3 µMC-12 carboxyl esterification requiredCHN[156]Antidiabeticpenicilliumin B (**131**)/fungusTerpenoid ^f^Glomerular mesangial cells fibrogenic inhibition0.5 µM *NADPH oxidase inhibitionCHN[157]Antidiabeticwailupemycin I (**132**)/bacteriumPolyketide ^d^α-glucosidase inhibition8.3 µMCompetitive inhibitionCHN[158]Antidiabeticasperentin B (**133**)/fungusPolyketide ^d^PTP1B inhibition2 μMUndeterminedDEU[159]Antidiabeticlasiodiplactone A (**134**)/fungusPolyketide ^d^α-glucosidase inhibition29.4 μMUndeterminedCHN[160]Antidiabetic7-hydroxy-de-O-methyllasiodiplodin (**135**)/bacteriumPolyketide ^d^α-glucosidase inhibition25.8 μMUndeterminedCHN[161]Antidiabeticsescandelin B (**136**)/fungusPolyketide ^d^
α-glucosidase inhibition17.2 μMUndeterminedCHN[162]Anti-inflammatoryAMT-E (**127**)/algaTerpenoid ^f^Murine colitis inhibition 10 mg/kg **Inhibition of TNF-α, IL-6ESP, MAR[163]Anti-inflammatory*Bacillus* sp. diketopiperazines (**138**–**140**)/bacteriumPeptide ^g^TGFBIp inhibition in vivo5 μM **Septic responses inhibitionS. KOR[164,165]Anti-inflammatory6-bromoisatin (**141**)/molluscAlkaloid ^g^Lung inflammation inhibition in vivo0.05 mg/g **Inhibition of TNF-α, IL-6AUS[166]Anti-inflammatoryceylonamide A (**142**)/spongeTerpenoid ^f^
Macrophage RANKL inhibition13 μM *SAR completedIDN, JPN, NLD[167]Anti-inflammatorycitrinin H1 (**143**)/fungusPolyketide ^e^Microglia NO and PGE_2_ release inhibition8 μMNF-κB inhibitionS. KOR, VNM[168]Anti-inflammatory nonenolide derivative (**144**)/algaPolyketide ^e^BMDC cytokine release inhibition7.6–10.9 μMJNK, ERK, AP-1, NF-κB inhibitionS. KOR[169]Anti-inflammatorycucumarioside A_2_-2 (**145**)/sea cucumberTerpenoid ^f^Binding of macrophage P2X purinergic receptors0.02 μM *Induction Ca^2+^ oscillationsRUS[170]Anti-inflammatorycurvularin derivative (**146**)/fungusPolyketide ^e^Macrophage PGE_2_ and NO release inhibition1.9–2.7 μMNF-κB signaling inhibition S. KOR[171]Anti-inflammatory9,11-dihydrogracilin A (**147**)/spongeTerpenoid ^f^PBMC proliferation inhibition3 μM *IL-6 and IL-10 inhibitionITA[172]Anti-inflammatorydysivillosin A (**148**)/spongeTerpenoid ^f^Basophil β-hexosaminidase inhibition8.2 μMIL-4 and LTB4 inhibitionCHN[173]Anti-inflammatoryepinecidin-1 (**149**)/fishPeptide ^g^Inhibition of MyD88 protein levels6 µg/mL *Proteasome degradation requiredTWN[174]Anti-inflammatoryexcavatolide B (**150**)/soft coralTerpenoid ^f^Attenuation of rat arthritis [175]2.5, 5 mg/kg **Decreased MMP-2, MMP-9, CD11b in tissuesCHN, TWN[175]Anti-inflammatoryfucoxanthin (**151**)/algaTerpenoid ^f^Decreased mice paw edema, adipogenesis and ear inflammation4 mg/kg **Modulation of iNOS, PLA_2_, COX-2, ACC, IL-6 and Nrf2 expression JPN, S. KOR, MEX[176,177,178,179]Anti-inflammatory*H. crispa* peptide (**152**)/sea anemonePeptide ^g^Macrophage histamine receptor inhibition10 μM *Intracellular Ca^2+^ increase inhibitionRUS[180]Anti-inflammatoryhipposponlachnin B (**153**)/spongeTerpenoid ^f^
Basophil β-hexosaminidase inhibition24 μMIL-4 and LTB4 inhibitionCHN[181]Anti-inflammatoryogipeptins A-D (**154**–**157**)/bacteriumPeptide ^g^Macrophage TNF-α production inhibition1 µM *Block LPS binding to CD14 JPN[182]Anti-inflammatoryoscarellin (**158**)/spongeAlkaloid ^g^Macrophage TNF-α and IL-6 expression inhibition>10 μMJNK, ERK, AP-1, NF-κB inhibitionS. KOR, USA[183]Anti-inflammatorypseudane-VIII (**159**)/bacteriumAlkaloid ^g^
Macrophage NO release inhibition6 µM *iNOS and IL-1β inhibitionS. KOR[184]Anti-inflammatoryacremeremophilane B (**160**)/fungusTerpenoid ^f^Macrophage NO release inhibition8 µMUndeterminedCHN, DEU[185]Anti-inflammatoryactinoquinolines A and B (**161**, **162**)/bacteriumAlkaloid ^g^COX-1 and -2 inhibition1.4–7.6 μMUndeterminedEGY, USA[186]Anti-inflammatoryanthenoside O (**163**)/starfishTerpenoid ^f^Macrophage SOX inhibition>10 µM *UndeterminedRUS, VNM[187]Anti-inflammatoryaurasperone C (**164**)/fungusPolyketide ^e^COX-2 inhibition4.2 μMUndeterminedCHN[188]Anti-inflammatorybriarenolides M and N (**165**, **166**)/soft coralTerpenoid ^f^Macrophage iNOS expression inhibition10 μM *UndeterminedTWN[189]Anti-inflammatorybriarenolides ZII and ZVI (**167**, **168**)/soft coralTerpenoid ^f^Macrophage iNOS and COX-2 expression inhibition10 μM *UndeterminedTWN[190]Anti-inflammatorydihydrobipolaroxin (**169**)/fungusTerpenoid ^f^Macrophage NO release inhibition>12.5 μM *UndeterminedCHN[191]Anti-inflammatoryechinulin (**170**)/fungusAlkaloid ^g^Microglia NO release inhibition4.6 μMUndeterminedCHN, S. KOR[192]Anti-inflammatory5α-iodozoanthenamine (**171**)/zoanthidAlkaloid ^g^Neutrophil SOX and elastase inhibition>10 μM *UndeterminedTWN[193]Anti-inflammatoryklyflaccisteroid J and K (**172**, **173**)/soft coralTerpenoid ^f^Neutrophil SOX and elastase inhibition1.5–5.8 μM UndeterminedTWN[194,195]Anti-inflammatory*L. varium* diterpenoid (**174**)/soft coralTerpenoid ^f^Neutrophil elastase inhibition>10 μM *UndeterminedEGY, SAU, TWN[196]Anti-inflammatorypetasitosterones B and C (**175**, **176**)/soft coralTerpenoid ^f^Neutrophil SOX and elastase inhibition2.7–4.4 µMUndeterminedTWN[197]Anti-inflammatory*Pinnnigorgia* sp. sterols (**177**, **178**)/soft coralTerpenoid ^f^Macrophage COX-2 and iNOS expression inhibition10 µM *UndeterminedTWN[198]Anti-inflammatorypinnigorgiol A (**179**)/soft coralTerpenoid ^f^Neutrophil SOX and elastase inhibition4, 5 µMUndeterminedTWN [199]Anti-inflammatorypinnigorgiol E (**180**)/soft coralTerpenoid ^f^Neutrophil SOX and elastase inhibition1.6, 3.9 µMUndeterminedTWN[200]Anti-inflammatorypinnisterols A and H (**181**, **182**)/soft coral Terpenoid ^f^Neutrophil SOX and elastase inhibition2.3–3.3 µMUndeterminedTWN[201,202] Anti-inflammatoryplancipyrroside B (**183**)/starfishTerpenoid ^f^Macrophage iNOS expression inhibition5.9 µMUndeterminedRUS, VNM[203]Anti-inflammatoryprotolinckioside A (**184**)/starfishTerpenoid ^f^Macrophage SOX inhibition10 µMUndeterminedIND, RUS[204]Anti-inflammatorysarcophytonolide O (**185**)/soft coralTerpenoid ^f^Macrophage iNOS expression inhibition8 μMUndeterminedCHN, USA[205]Anti-inflammatorysinularectols A and B (**186**, **187**)/soft coralTerpenoid ^f^Neutrophil SOX and elastase inhibition0.9–8.5 μMUndeterminedTWN[206]Anti-inflammatorysinubrasolides A and D (**188**, **189**)/soft coralTerpenoid ^f^Neutrophil SOX and elastase inhibition1.4–8 μMUndeterminedCAN, SAU, TWN[207,208]Anti-inflammatoryuprolides N, O and P (**190**–**192**)/soft coralTerpenoid ^f^Macrophage TNF-α and IL-6 release inhibition1.4–4.2 μMUndeterminedIND, PAN, USA[209]Immune systemcucumarioside A_2_-2 (**145**)/sea cucumberTerpenoid ^f^Increase in spleen white pulp and macrophage activation3 mg/kg **Increased B cell PCNA and M1 macrophages RUS, TWN[210,211]Immune systemgracilins A, H and L(**193**–**195**)/spongeTerpenoid ^f^CD147 receptor modulation and T-cell IL-2 release inhibition1 μM *Hypersensitivity and NFATc inhibitionESP, GBR[212,213]Immune systemshinorine and porphyra-334 (**196**, **197**)/algaPeptide ^g^NF-κB stimulation50 µg/mL *Tryptophan metabolism modulationAUT[214]Immune systemsinulariolide (**198**)/soft coralTerpenoid ^f^Dendritic cell maturation suppression25 µg/mL *IL-6, IL-12 and NO inhibitionTWN[215]Immune systemCDMW-3 (**199**)/fungusPeptide ^g^PCA inhibition in vivo20 mg/kg **Mast cell histamine and cytokine release inhibitionCHN[216]Immune systemchrysamide C (**200**)/fungusAlkaloid ^g^IL-17 inhibition>1 μM *UndeterminedCHN[217]Immune systemcocosolide (**201**)/cyanobacteriumPolyketide ^d^IL-2 inhibition2.5 μM *UndeterminedCHN, USA[218]Immune systemmyxillin A (**202**)/spongeAlkaloid ^g^IL-12p40 release inhibition10 µg/mL *UndeterminedDNK, ISL[219]Immune systempectinioside A (**203**)/starfishTerpenoid ^f^Increase OVA-specific IgG1 in vivo25 µg *UndeterminedJPN[220]Immune systempeniphenone (**204**)/fungusPolyketide ^d^Lymphocyte immune suppression8.1–9.3 µg/mLUndeterminedCHN[221]Immune systemUSF-19A (**205**)/bacteriumPeptide ^g^Splenocyte IL-5 release inhibition0.57 μM UndeterminedCHN[222]Nervous systemAPETx4 (**206**)/sea anemonePeptide ^g^Kv10.1 potassium channel inhibition1.1 µMBinds channel in closed stateBEL, DEU [223]Nervous systemastaxanthin (**207**)/shrimpTerpenoid ^f^Penitrem A toxicity reversal 20 µM *Block BK channel EGY, USA[224]Nervous systemcrambescidin 816 (**208**)/spongeAlkaloid ^g^Cortical neurons cytosolic Ca^2+^ increase10 µM *AMPA and NMDA receptors involvedESP, FRA, IRL[225]Nervous system*C. generalis* O-conotoxin (**209**)/cone snailPeptide ^g^Α9α10 nACh receptor inhibition16.2 nMNon-competitive inhibitionAUS, CHN[226]Nervous system*C. princeps* PiVIIA peptide (**210**)/cone snailPeptide ^g^Neuronal Ca^2+^ current increase3 µM *Potentiates two types Ca^2+^ channelsCUB, MEX[227]Nervous systemconorphin T (**211**)/cone snailPeptide ^g^KOR agonist9.8 µM*In vivo* colonic receptor inhibitionAUS[228]Nervous system11-dehydrosinulariolide (**212**)/soft coralTerpenoid ^f^Amelioration PD and spinal cord injury attenuation 5 µg/rat **DJ-1 expression upregulation and microglia activationTWN[229,230]Nervous systemdiscorhabdin G (**213**)/spongeAlkaloid ^g^
Eel and human AChE inhibition1.3 µMReversible competitive inhibitionDEU, ITA, SVN[231]Nervous systemfucoxanthin (**151**)/algaTerpenoid ^f^BACE1 inhibition5.3 µMMixed inhibitionGBR, S. KOR[232]Nervous systemfucoxanthin (**151**)/algaTerpenoid ^f^Reversal BDNF expression50 mg/kg **Reversed AChE activityCHN[233]Nervous systemfucoxanthin (**151**)/algaTerpenoid ^f^Neuroprotection after TBI-induced brain injury100 mg/kg **Nrf2-ARE pathway modulationCHN[234]Nervous system5-hydroxycyclopenicillone (**214**)/fungusPolyketide ^d^H_2_O_2_-induced neuronal death protection30 µM *DHHP free radical inhibitionCHN, USA[235] Nervous systemmaitotoxin (**215**)/alga Polyketide ^d^Activation of NSCC 10 pM *TRPC1 inhibitionMEX[236]Nervous systemmakaluvamine J (**216**)/spongeAlkaloid ^g^Reduction of mitochondrial damage0.1–1 µM *Nrf2 activationESP, FJI, GBR[237]Nervous systemMEC-1 (**217**)/spongePolyketide ^d^AChE inhibition20.9 µMDocking studiesEGY[238]Nervous systemmellpaladine A (**218**)/ascidianAlkaloid ^g^*In vivo* behavior modulation8 nM/mouse **Serotonin receptor affinityJPN[239]Nervous systemMs 9a-1 peptide (**219**)/sea anemonePeptide ^g^Decrease in nociceptive and inflammatory response in vivo0.3 mg/kg **TRPA1 modulationNOR, RUS[240]Nervous tissuephlorofucofuroeckol-A (**220**)/algaPolyketide ^d^Glutamate-induced neurotoxicity inhibition10 µM *Intracellular and mitochondrial ROS inhibtionS. KOR[241]Nervous systempiloquinone (**221**)/bacteriumPolyketide ^d^MAO-B inhibition1.2 µMReversible competitive inhibitionS. KOR, USA[242]Nervous systempseudopterosin A (**222**)/soft coralTerpenoid ^f^Synaptic transmission alteration1 µM *Extensive brain distribution USA[243]Nervous systemsqualamine (**223**)/sharkTerpenoid ^f^Reduction of α-synuclein aggregation in vivo50 µM **α-synuclein displaced from lipid membranesESP, GBR, ITA, NLD, USA[244]Nervous systemstryphnusin (**224**)/spongeAlkaloid ^g^Eel AChR inhibition232 µMReversible competitive inhibitionHRV, NOR, SVN, SWE[245]Nervous systemxyloketal B (**225**)/fungusPolyketide ^d^Cerebral infarction modulation50 mg/kg **Decreased ROS and cytokinesCAN, CHN, USA[246]Nervous systemaraplysillin X (**226**)/spongeAlkaloid ^g^BACE1 inhibition31.4 µMUndeterminedNZL, USA[247]Nervous systemcaracolamide A (**227**)/cyanobacterium Alkaloid ^g^Ca^2+^ channel modulation10 pM *UndeterminedBRA, JOR, PAN, USA, [248]Nervous systemconorfamide-Sr3 (**228**)/snailPeptide ^g^Blocks volatage-gated K^+^ channel2.7 µM ****Shaker* channel specificMEX[249]Nervous systemcontryphan-Bt (**229**)/cone snailPeptide ^g^Stiff-tail syndrome in vivo5 ng/mouse **UndeterminedCHN[250]Nervous systemdehydroaustin (**230**)/fungusMeroterpenoid ^f^AChE inhibition0.4 µMUndeterminedCHN[251] Nervous systemhymenidin (**231**)/spongeAlkaloid ^g^K_v_1.3- K_v_1.6 K+ channel inhibition2.5–7.6 µMUndeterminedBEL, GBR, SVN[252]Nervous systempsammaplysene A (**232**)/spongeAlkaloid ^g^Binding to RNA-binding protein HNRNPK86.2 µM ***UndeterminedUSA[253]Nervous systemterreulactone C (**233**)/fungusMeroterpenoid ^f^AChE inhibition28 nMUndeterminedCHN[254]Nervous systemturripeptide (**234**)/turrid snailPeptide ^g^Α9α10 nAChR inhibition10.2 µMUndeterminedAUS, KAS, MEX, PHL, USA[255]**^a^ Organism:***Kingdom Animalia*: shrimp (Phylum Arthropoda); ascidian, fish (Phylum Chordata; coral, sea anemone and zoanthid (Phylum Cnidaria); sea cucumber, starfish (Phylum Echinodermata); cone snail, turrid snail (Phylum Mollusca); sponge (Phylum Porifera); *Kingdom Fungi*: fungus; *Kingdom Plantae:* alga; diatoms; *Kingdom Monera*: bacterium; **^b^ IC_50_**: concentration of a compound required for 50% inhibition, *: apparent IC_50_, ** in vivo study; ***: *K*_i_: concentration needed to reduce the activity of an enzyme by half; **^c^ MMOA:** molecular mechanism of action; **^d^ Country:** AUS: Australia; AUT: Austria; BEL: Belgium; BRA: Brazil; CAN: Canada; CHN: China; CUB: Cuba; DEU: Germany; DNK: Denmark; EGY: Egypt; ESP: Spain; FJI: Fiji; FRA: France; GBR: United Kingdom; HRV: Croatia; IDN: Indonesia; IND, India; ISL: Iceland; IRL: Ireland; ITA: Italy; JOR: Jordan; JPN: Japan; KAS: Kazakhstan; MAR: Morocco; MEX: Mexico; NZL: New Zealand; NLD: Netherlands; NOR: Norway; PAN: Panama; PHL: Philippines; RUS: Russian Federation; SAU: Saudi Arabia; S. KOR: South Korea; SVN: Slovenia; SWE: Sweden; TWN: Taiwan; VNM: Vietnam; **Chemistry: ^e^** Polyketide; **^f^** Terpene; **^g^** Nitrogen-containing compound; **^h^** Shikimate. **Abbreviations:** ACC: acetyl-CoA carboxylase; Ach: acetylcholine; AChE: acetylcholinesterase; Akt: also known as protein kinase B is a serine/threonine protein kinase; AMPA: *α*-amino-3-hydroxy-5-methyl-4-isoxazolepropionic acid; AMPK: AMP-activated protein kinase; AMT-E: 11-hydroxy-1′-*O*-methylamentadione; AP-1: activator protein-1; BACE1: beta secretase aspartic protease; BDDE: Bis (2,3-dibromo-4,5-dihydroxybenzyl) ether; BK: voltage-gated potassium channels; BMDC: bone marrow-derived dendritic cells; CD14: cluster of differentiation 14; COX: cyclooxygenase; DHHP: α, α-diphenyl-β-picrylhydrazyl; DJ-1: cysteine protease encoded by PARK7 gene; ERK: extracellular signal-regulated kinase; HNRNPK: heterogenous nuclear ribonucleoprotein K; IL: interleukin; iNOS: inducible nitric oxide synthase; JNK: c-Jun NH_2_-terminal kinase; LPS: Lipopolysaccharide; MAO-B: monoamine oxidase B; MyD88: myeloid differentiation primary response protein 88; iNOS: inducible nitric oxide synthase; KOR: κ-opioid receptor; Kv current: voltage-gated K+ current; nAChR: nicotinic acetylcholine receptor; NF-κB: nuclear factor kappa-light-chain-enhancer of activated B cells; NMDA: *N*-methyl-d-aspartate receptor; NO: nitric oxide; nAChR: nicotinic acetylcholine receptor; Nrf2-ARE: nuclear transcription factor E2-related factor antioxidant response element; NSCC: non-selective cation channel; PBMC: human peripheral blood mononuclear cells; PCA: passive cutaneous anaphylaxis; PCNA: proliferating cell nuclear antigen; PD: Parkinson’s disease; PKB: protein kinase B; PLA_2_: phospholipase A_2_; PTP1B: tyrosine protein; phosphatase 1B; RANKL: receptor activator of nuclear factor-κB ligand; ROS: reactive oxygen species; SOX: superoxide; SQDC: sulfoquinovosyl diacylglycerols; TGFBIp: transforming growth factor β- induced protein; TNF-α: tumor necrosis factor-α; TRPA1: transient receptor potential ankyrin-repeat 1 receptor; TRPC1: transient receptor potential canonical type 1.



Figure 2Marine pharmacology in 2016–2017: marine compounds with antidiabetic and anti-inflammatory activity; and affecting the immune and nervous system.
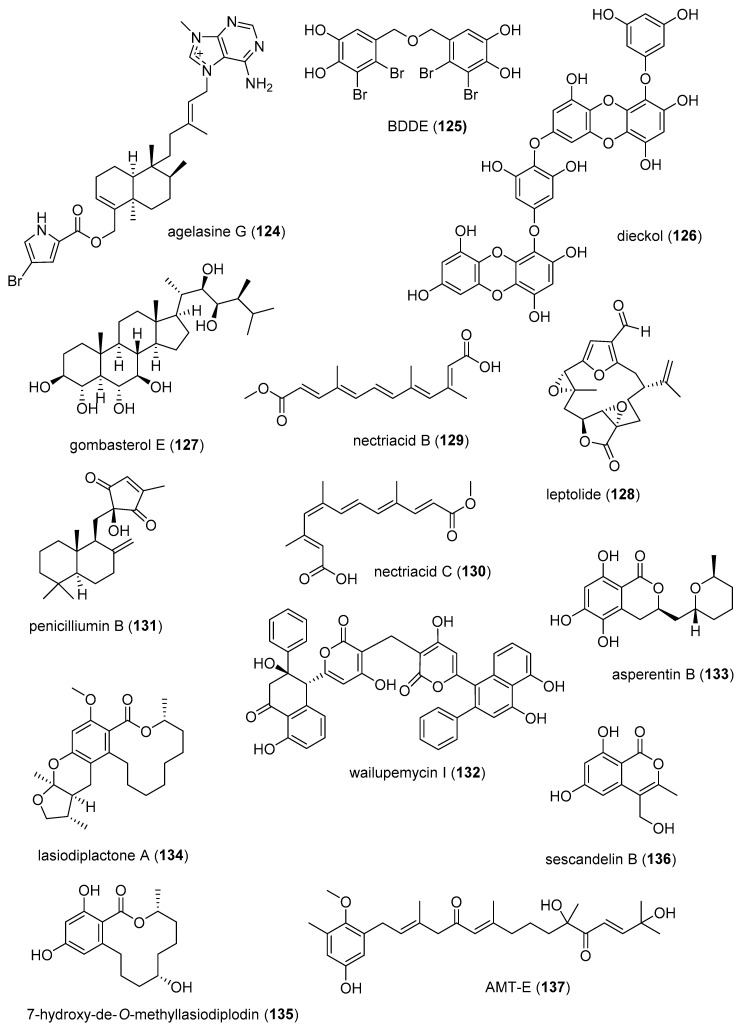

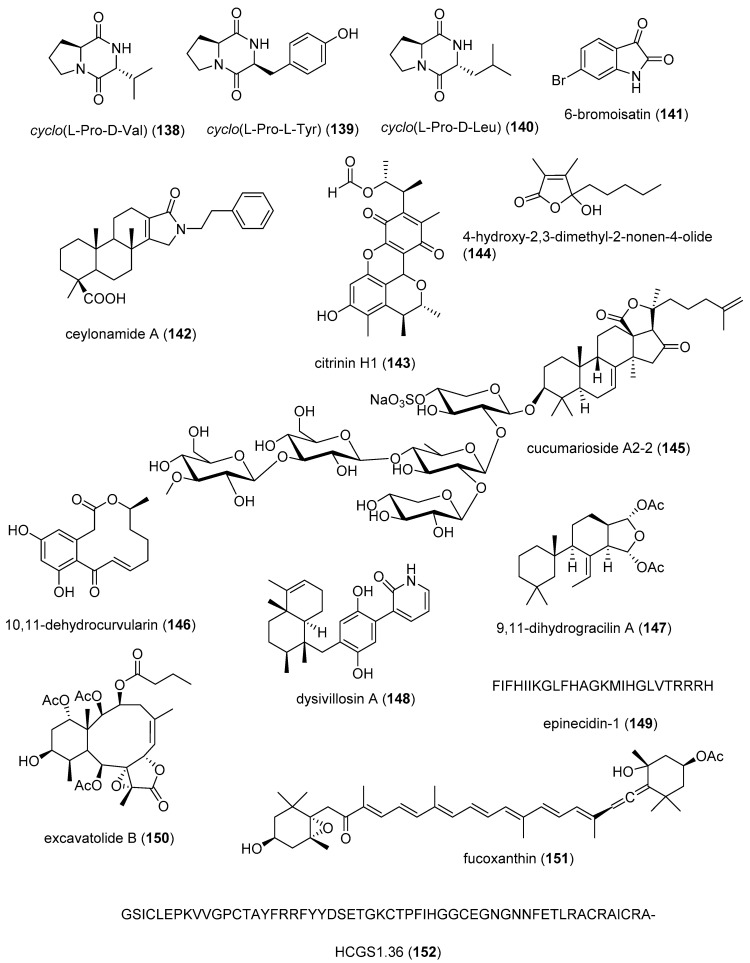

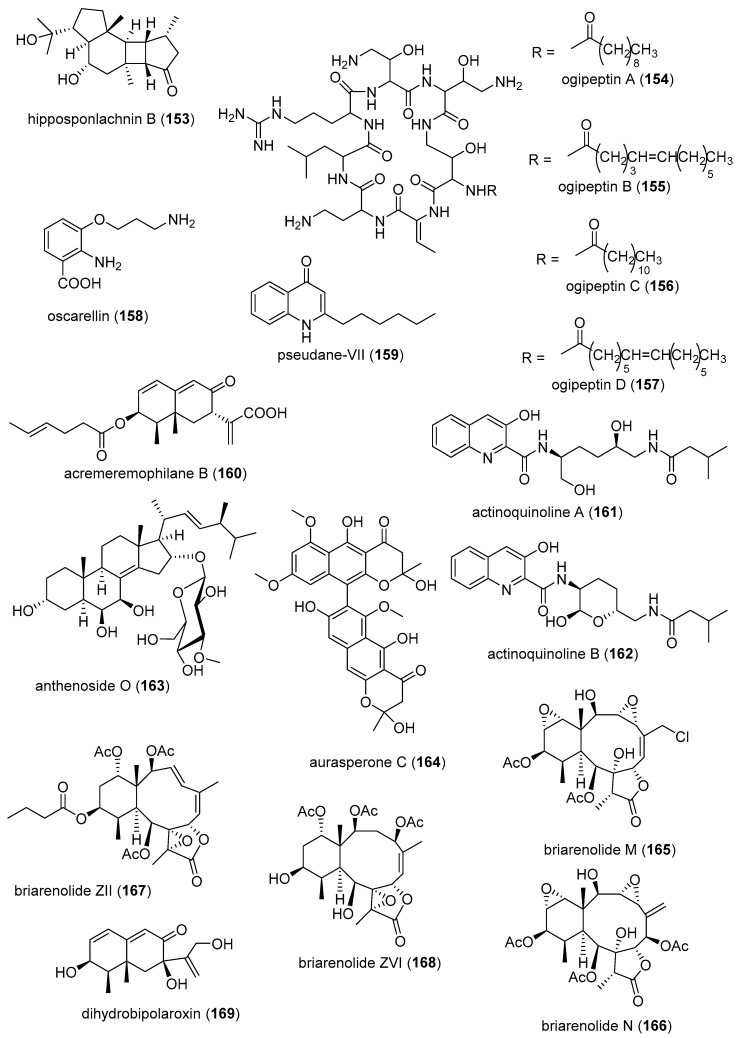

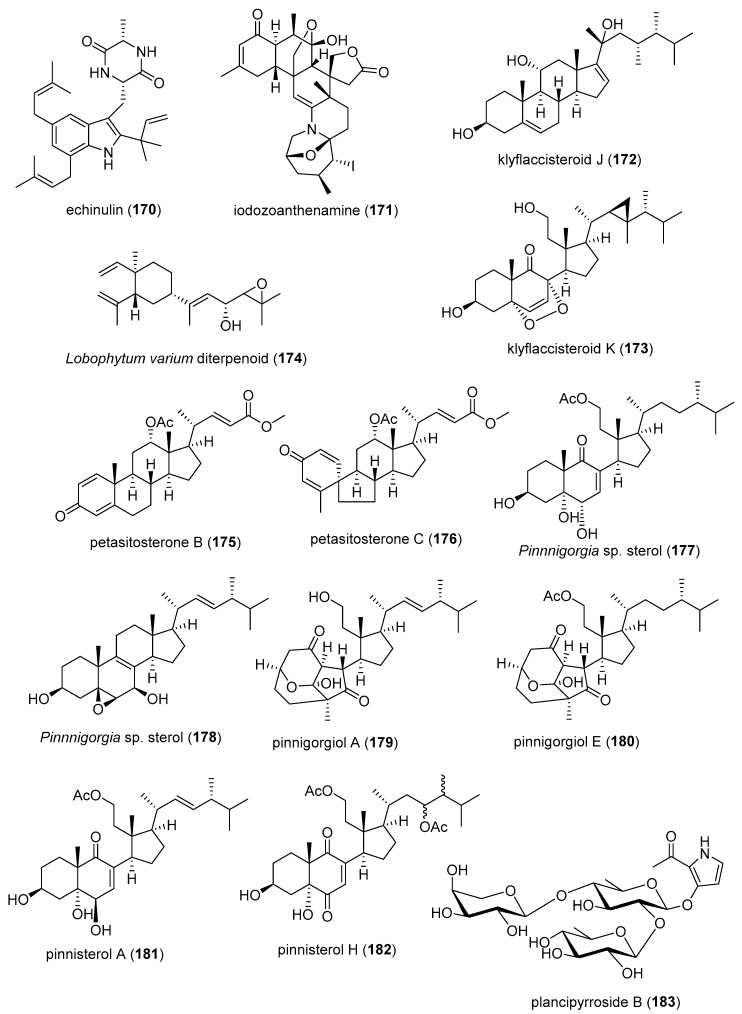

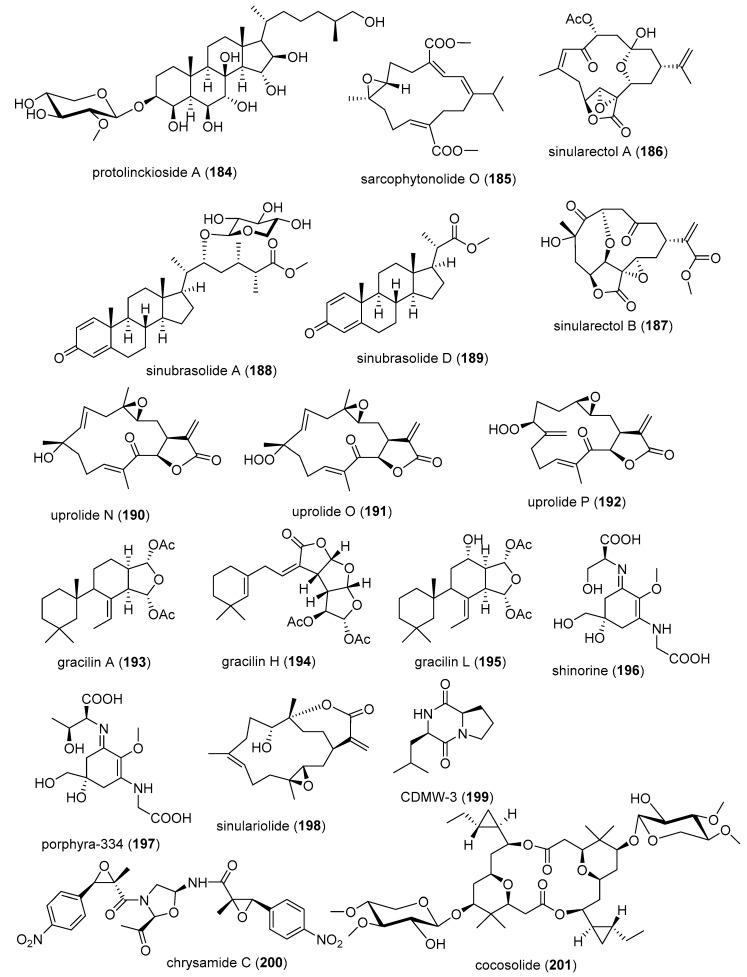

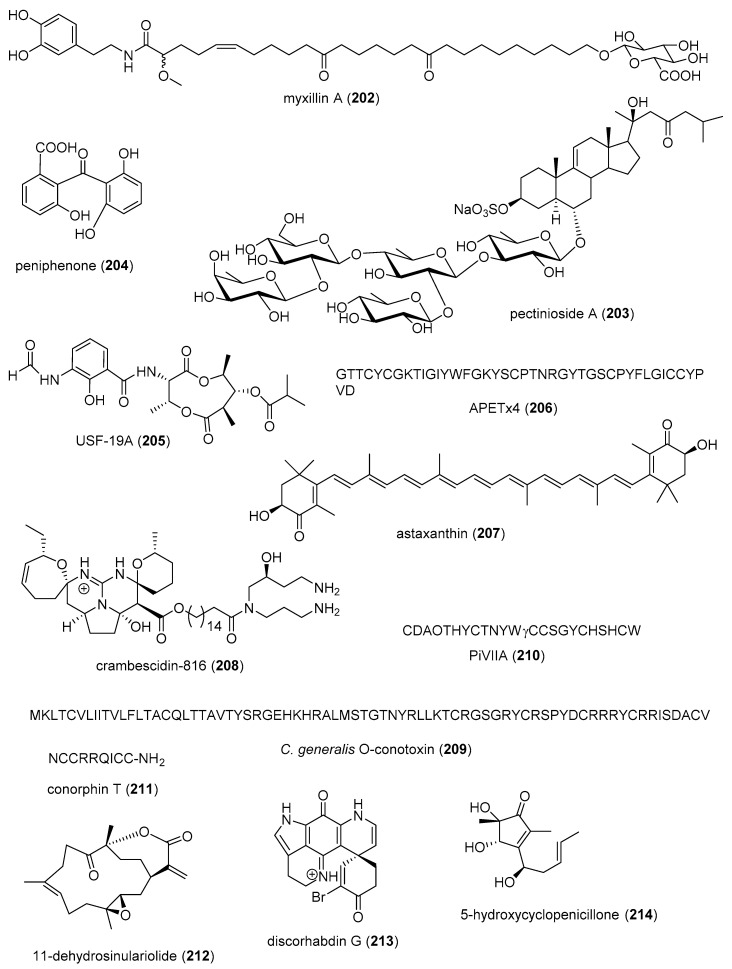

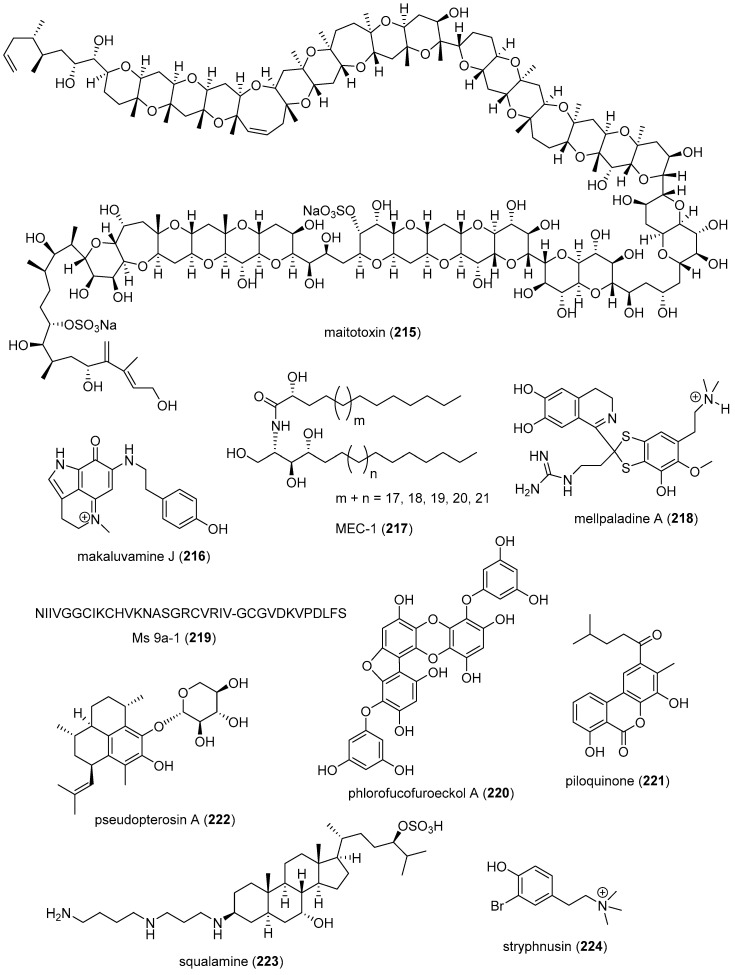

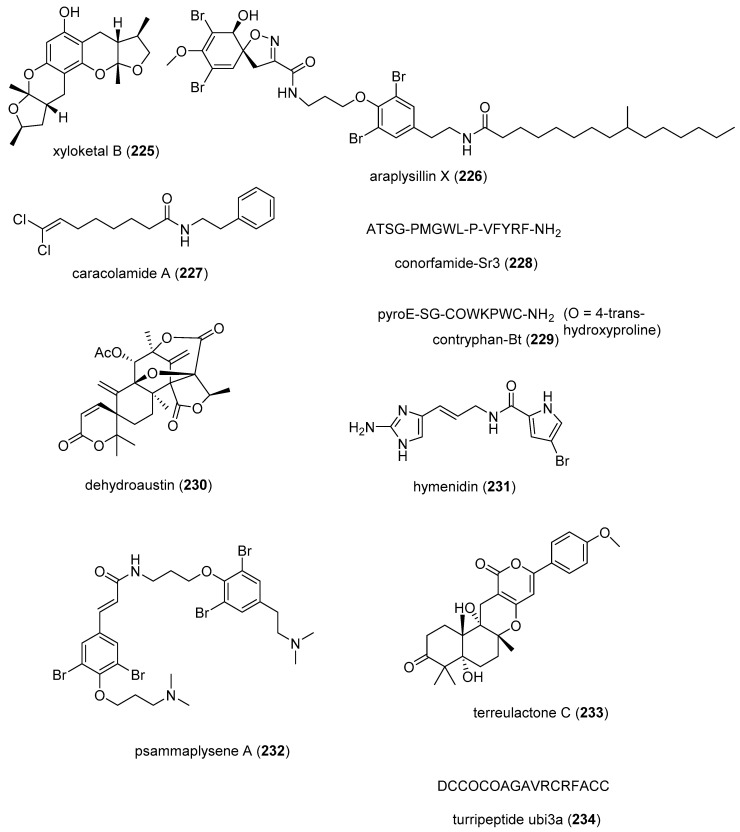



### 3.1. Antidiabetic Activity

As shown in Table 2 and Figure 2, twelve publications reported on the mode of action of marine-derived antidiabetic compounds (**124**–**132**) during 2016–2017. Yamazaki and colleagues contributed to the pharmacology of diabetes by noting that the diterpene marine alkaloid agelasine G (**124**) isolated from the Japanese marine sponge *Agelas nakamurai* selectively inhibited protein tyrosine phosphase B (PTP1B) and enhanced insulin-stimulated phosphorylation of serine/threonine protein kinase B or Akt in vitro, noting that further studies may “provide a candidate for anti-diabetes therapeutic agents” [151]. Xu and colleagues observed that a novel bromophenol bis (2,3-dibromo-4,5-dihydroxybenzyl) ether (BDDE) (**125**) isolated from the red alga *Odonthalia corymbifera*, increased glucose uptake in vitro, decreased the expression of protein tyrosine phosphatase 1B, activated the insulin signaling pathway and in mice “significantly decreased the blood glucose”, thus, suggesting BDDE might be a “treatment of type-2 diabetes” [152]. Kim and colleagues reported that the marine algal polyphenol dieckol (**126**), isolated from the marine brown alga *Ecklonia cava* attenuated blood glucose levels in the zebrafish model of hyperglycemia, an in vivo paradigm used to study chronic diseases, such as diabetes, by a mechanism that stimulated the protein kinase B (Akt) pathway, thus demonstrating the “anti-diabetic effect” of this compound [153]. Woo and colleagues showed that the novel polyoxygenated steroid gombasterol E (**127**) isolated from the Korean marine sponge *Clathria gombawuiensis* moderately enhanced both glucose uptake in adipocytes and phosphorylation of AMP-activated protein kinase and acetyl-CoA carboxylase in mouse skeletal myoblasts [154]. Villa-Pérez and colleagues reported that the furanocembranolide diterpenoid leptolide (**128**) isolated from soft coral *Leptogorgia alba* improved insulin sensitivity by increasing intracellular insulin signaling in both liver and skeletal muscle tissues of a diet-induced obese mice model, a “preclinical model of insulin resistance”, concluding that “furanocembranolides as a new therapeutic class to treat Type 2 diabetes” [155]. Cui and colleagues demonstrated that two new polyketides nectriacids B and C (**129**, **130**) isolated from a South China Sea mangrove *Sonneratia ovata*-derived endophytic fungus *Nectria* sp. HN001 moderately inhibited α-glucosidase, a significant finding because this enzyme prevents “breaking down complex carbohydrates for absorption” [156]. Lin and colleagues discovered a new sesquiterpene penicilliumin B (**131**) isolated from South China Sea deep sea (1300 m) sediment-derived *Penicillium* strain F00120 that potently inhibited kidney fibrogenic action of high glucose in vitro through oxidative stress disruption, thus, suggesting this compound had potential for “therapy of diabetic nephropathy” [157]. Chen and colleagues studied a polyketide wailupemycin I (**132**) isolated from a Chinese marine alga *Enteromorpha prolifera*-derived Streptomyces sp. OUCMDZ-3434 that moderately inhibited α-glucosidase by competitive inhibition of the enzyme [158].

Moreover, four marine natural products (**133**–**136**) listed in Table 2 and shown Figure 2, demonstrated antidiabetic activity during 2016–2017, but the mechanism of action of these compounds remained undetermined at the time of publication: a new polyketide asperentin B (**133**) isolated from a deep (2769 m) Mediterranean Sea sediment-derived *Aspergillus sydowii* which “strongly” inhibited human protein tyrosine phosphatase 1B, an important “target for the treatment of type 2 diabetes” [159]; a novel lactone lasiodiplactone (**134**) isolated from a South China Sea mangrove *Acanthus ilicifolius*-derived endophytic fungus *Lasiodiplodia theobromae* ZJ-HQ1 that exhibited α-glucosidase inhibitory activity that was better “than the clinical α-glucosidase inhibitor acarbose” [160]; a new de-*O*-methyllasiodiplodin (**135**) isolated from the co-cultivation of the Chinese mangrove *Clerodendrum inerme*-derived endophytic fungus *Trichoderma* sp. 307 and the aquatic pathogenic bacterium *Acinetobacter johnsonii* B2 that inhibited α-glucosidase better “than the positive control acarbose” [161]; and a known isocoumarin analogue sescandelin B (**136**) isolated form the Chinese *Kandelia obovata*-derived endophytic fungus *Talaromyces amesolkiae,* which showed moderate inhibition of α-glucosidase, “which was much better than acarbose” [162].

### 3.2. Anti-Inflammatory Activity

As shown in Table 2 and Figure 2, there was a remarkable increase in anti-inflammatory pharmacology of marine compounds (**127**, **138**–**192**) during 2016–2017. The molecular mechanism of action of anti-inflammatory marine natural products (**127**, **138**–**159**) was assessed in both in vitro and in vivo preclinical pharmacological studies in twenty one papers, which used several in vitro and in vivo models of inflammation.

Zbakh and colleagues evaluated the anti-inflammatory properties of the meroterpene 11-hydroxy-1′-*O*-methylamentadione (AMT-E) (**127**) isolated from the brown alga *Cystoseira usneoides* in a murine model of experimental colitis, observing that the levels of myeloperoxidase, cytokines and the expression of the pro-inflammatory genes nitric oxide synthase (iNOS) and cyclooxygenase-2 (COX-2) were reduced, concluding that AMG-W might be a candidate for “prevention/treatment of inflammatory bowel disease” [163]. Jung and colleagues investigated three known diketopiperazines (**138**–**140**) isolated from a Korean marine sediment-derived bacteria *Bacillus* sp. HC001 and *Piscicoccus* sp. 12L081 and observed that they effectively inhibited LPS-induced release and expression of transforming growth factor β-induced protein in human endothelial cells, as well as protected against cecal ligation and puncture-induced lethality and organ damage in a murine model of experimental sepsis [164,165]. Ahmad and colleagues reported that the orally administered brominated indol 6-bromoisatin (**141**) isolated form the Australian marine gastropod mollusc *Dicathais orbita* inhibited inflammation in a murine model of lipopolysaccharide-induced acute lung injury by significantly reducing pro-inflammatory tumor necrosis-α (TNF-α) and interleukin-1 β (IL-1β) production and associated lung damage, thus, concluding this compound “provides a lead for the development of safer anti-inflammatory drugs” [166]. El-Desoky and colleagues studied a new nitrogenous diterpene ceylonamide A (**142**), isolated from the Indonesian marine sponge *Spongia ceylonensis* and determined that they exhibited moderate in vitro anti-inflammatory activity of receptor activator of nuclear factor-κB ligand (RANKL)-induced osteoclastogenesis in RAW264 macrophages with structure-activity relationships studies. This revealed that the “position of the carbonyl group of the γ-lactam ring …at its nitrogen atom were important for inhibitory activity” [167]. Ngan and colleagues isolated a known polyketide citrinin H1 (**143**) from the Korean sand-derived bacterium *Penicillium* sp. SF-5629 and observed moderate inhibition *E. coli* LPS-activated BV-2 microglia in vitro nitric oxide (NO) and prostaglandin E_2_ (PGE_2_) release with concomitant downregulation of the inducible enzymes nitric oxide synthase and cyclooxygenase-2 attributed to “inhibition of NF-κB and p38 MAPK signaling pathways” [168]. Manzoor and colleagues observed that 4-hydroxy-2,3-dimethyl-2-nonen-4-olide (**144**) extracted from Korean marine alga *Ulva pertusa* moderately inhibited pro-inflammatory cytokines IL-12 p40 and IL-6 release from bone marrow-derived dendritic cells, as well as signal transduction by inhibiting phosphorylation of nuclear factor -κB (NF-κB), and thus, warranted further study to evaluate its potential as a “therapeutic agent for inflammation-associated maladies” [169]. Aminin and colleagues assessed the anti-inflammatory pharmacology of the triterpene glycoside cucumarioside A_2_-2 (**145**) isolated from the Russian edible sea cucumber *Cucumaria japonica* by demonstrating that this compound potently interacted with P2X purinergic receptors, in particular P2X4, on mature macrophage membranes and enhanced reversible ATP-dependent Ca^2+^ signaling thus leading to “activation of cellular immunity” [170]. Ha and colleagues studied the anti-inflammatory effects of a known curvularin-type metabolite (**146**) derived from an Antarctic Ross Sea sponge-derived fungus *Penicillium* sp. SF-5850 that potently inhibited LPS-induced RAW264.7 macrophages production of pro-inflammatory NO and PGE_2_ as well as the expression of iNOS and COX-2 by inhibition of the NF-κB, a “transcriptional factor(s) involved in inflammation-related disorders” [171]. Ciaglia and colleagues characterized the anti-inflammatory effects of the terpenoid 9,11-dihydrogracilin A (**147**) isolated from the Antarctic marine sponge *Dendrilla membranosa* and observed that, in human peripheral blood mononuclear cells, in vitro, it downregulated NF-κB as well as inhibition of IL-10, while in vivo it significantly attenuated mouse ear edema and inflammation, thus supporting the use of this compound “in inflammatory cutaneous diseases [172]. Jiao and colleagues investigated the “terpene-polyketide-pyridine hybrid” dysivillosin A (**148**) isolated from the South China Sea marine sponge *Dysidea villosa* that inhibited RBL-2H3 mast cell release of β-hexosaminidase, a marker of degranulation, as well as pro-inflammatory leukotriene B4 and IL-4 by suppressing the IgE/Syk signaling pathway, thus suggesting a “new chemotherapeutic scaffold targeting Syk-associated allergy” [173]. Su and colleagues extended the pharmacology of the known antimicrobial peptide epinecidin-1 (**149**) isolated from the orange-spotted grouper *Epinephelus coioides* by showing that its anti-inflamatory activity resulted from degradation of the Toll-like receptor signaling adaptor protein MyD88 in RAW 264.7 mouse macrophages and concomitant activation of the Smurf E3 ligase proteasome degradation pathway [174]. Lin and colleagues continued the evaluation of the terpenoid excavatolide (**150**) isolated from the Taiwanese gorgonia coral *Briareum excavatum,* observing that it inhibited LPS-induced osteoclast-like cell formation and tartrate-resistant acid phosphatase (TRAP) expression, in vitro, while also significantly reducing paw oedema and TRAP-positive multinucleated cell formation in two rat models of experimental arthritis [175]. Several studies extended the anti-inflammatory mechanisms of the marine carotenoid fucoxanthin (**151**) isolated from the edible brown alga *Undaria pinnatifida*: Choe and colleagues reported that fucoxanthin inhibited paw edema in an experimental in vivo model of inflammation by reducing activation of iNOS, COX-2 and NF-κB [176]; Grassa-López and colleagues observed that fucoxanthin ameliorated lipogenesis, decreased insulin resistance as well as biomarkers of both inflammation and cardiovascular disfunction in an experimental rat obesity model [177]; Sugiura and colleagues determined that in vitro fucoxanthin suppressed PLA_2_ and COX-2 expression in a rat basophilic leukemia-2H3 cells, while in vivo inhibiting PLA_2_, COX-2 and hyaluronidase in several ICR mouse ear models of inflammation [178]; Taira and colleagues observed that concentration-dependent cytoprotection or apoptosis in RAW264.7 macrophage cells by the carotenoids fucoxanthinol and fucoxanthin resulted from activation of the Nrf2-ARE signaling pathway [179]. Sintsova and colleagues characterized a novel Kunitz-type peptide (**152**) isolated from the sea anemone *Heteractis crispa* that significantly decreased intracellular Ca^2+^ in histamine-treated murine bone marrow-derived macrophages thus suggesting the involvement of “H_1_-type histamine receptor blockage” in the anti-inflammatory mechanism of action of this peptide [180]. Hong and colleagues determined that the diterpenoid hipposponlachnin B (**153**) isolated from the South China Sea marine sponge *Hippospongia lachne* decreased production of β-hexosaminidase, a degranulation biomarker and pro-inflammatory mediators IL-4 and LTB4 by RBL-2H3 cells, suggesting a possible therapeutic use for “the treatment of allergy” [181]. Kozuma and colleagues isolated new cyclic peptides ogipeptins A-D (**154**–**157**) from the culture broth of the Japanese marine Gram-negative bacterium *Pseudoalteromonas* sp. SANK 71903 that blocked the binding of LPS to the cluster of differentiation 14 (CD14) in vitro and decreased TNF-α release by human U937 monocytic cells, concluding that these peptides could be developed for use “as anti-LPS drugs against LPS-associated diseases” [182]. Kwon and colleagues reported a new anthranilic acid derivative oscarellin (**158**) isolated from a Philippine sponge *Oscarella stillans* that strongly inhibited LPS-induced TNF-αand IL-6 production in murine macrophage RAW 264.7 macrophages by a mechanism “associated with inactivation of c-Jun NH_2_-terminal kinase (JNK), extracellular signal-regulated kinase (ERK), activator protein-1 (AP-1), and NF-κB and activation of activating transcription factor-3 (ATF-3)” [183]. Kim and colleagues observed that the novel alkaloid pseudane-VII (**159**) isolated from the marine bacterium *Pseudolateromonas* sp. M2 modulated release of pro-inflammatory mediators (NO, IL-1β and IL-6) by LPS-treated RAW 264.7 murine macrophage cell line in vitro via inhibition of “MAPK phosphorylation and NF-κB translocation”, while in vivo treatment with this compound suppressed production of IL-1β and iNOS expression, suggesting “pseudane VII has potential as treatment for inflammatory responses and diseases” [184].

In contrast, for marine compounds (**160**–**192**) shown in Table 2 and Figure 2, only the anti-inflammatory activity (IC_50_) was reported, no molecular mechanism of action was reported at the time of publication: A new eremophilane-sesquiterpene acremeremophilane B (**160**) isolated from a South Atlantic deep sea (2869 m)-sediment derived fungus *Acremonium* sp. [185]; two new quinoline alkaloids actinoquinolines A and B (**161**, **162**) isolated from a Californian marine-derived *Streptomyces* sp. strain CNP975 [186]; a new polyhydroxysteroidal glycoside anthenoside O (**163**) isolated from the tropical South China Sea starfish *Anthenea aspera* [187]; a new polyketide aurasperone C (**164**), isolated from a South China Sea brown alga *Sargassum* sp.-derived *Aspergillus niger* SCSIO Jcsw6F30 [188]; two new 9-hydroxybriarane diterpenoids briarenolides ZII and ZVI (**167**, **168**) isolated from the Taiwanese gorgonian coral *Briareum* sp. [190]; a known eremophilane sesquiterpene dihydrobipolaroxin (**169**) isolated from a South China Sea deep (2439 m) marine sediment-derived *Aspergillus* sp. SCSIOW2 [191]; a known alkaloid echinulin (**170**) isolated from an Antarctic Ross Sea “unidentified” marine organism-derived *Aspergillus* sp. SF-5976 [192]; a novel halogenated 5α-iodozoanthenamine (**171**) isolated from the colorful Taiwanese zoanthid *Zoanthus kuroshio* [193]; two novel steroids klyflaccisteroid J and K (**172**, **173**) isolated from the Taiwanese soft coral *Klyxum flaccidum* [194,195]; a known lobane diterpenoid (**174**) isolated from a Taiwanese soft coral *Lobophytum varium* [196]; two new bioactive steroids petasitosterones B and C (**175**, **176**) isolated from the Taiwanese soft coral *Umbellulifera petasites* [197]; two new sterols (**177**, **178**) [198], and novel 9,11-secosterols pinnigorgiol A (**179**) [199], pinnigorgiol E (**180**) [200], pinnisterol A (**181**) [201] and pinnisterol H (**182**) [202] isolated from the Taiwanese gorgonian coral *Pinnigorgia* sp.; a new pyrrole oligoglycoside plancipyrroside B (**183**) isolated from an extract of the Vietnamese starfish *Acanthaster planci* [203]; a novel steroidal glycoside protolinckioside A (**184**) isolated from an Arabian sea starfish *Protoreaster lincki* [204]; a known terpenoid sarcophytonolide O (**185**) isolated from the Chinese soft coral *Lobophytum crassum* [205]; two novel norcembranoids sinulerectols A and B (**186**, **187**) isolated from South China Sea soft coral *Sinularia erecta* [206]; a known terpenoid sinubrasolide A (**188**) and the steroid sinubrasone D (**189**) isolated from the Taiwanese “cultivation pool” soft coral *Sinularia brassica* [207,208]; and the novel diterpenes uprolides N, O and P (**190**–**192**) isolated from the Panamanian octocoral *Eunicea succinea* [209].

### 3.3. Marine Compounds with Activity on the Immune System

In 2016–2017, the preclinical pharmacology of marine compounds that were reported to affect the immune system is shown in Table 2 and Figure 2. The molecular mechanism of action of marine natural products that affected the immune system (**145**, **193**–**199)** was described in seven papers which used several in vitro and in vivo models. Pislyagin and colleagues reported that the triterpene glycoside cucumarioside A_2_-2 (**145**) isolated from the Far Eastern sea cucumber *Cucumaria japonica* caused changes in mouse spleen morphology, proliferative activity “predominantly in B-cells” and macrophage activation in mouse spleen, concomitant with increase in IL-1β, iNOs, ROS and NO production [210,211]. Sánchez and colleagues investigated three terpenoids gracillin A, H and L (**193**–**195**) isolated from the marine sponge *Spongionella gracillis* that inhibited human T lymphocyte IL-2 production, as well as CD147 expression by reducing calcineurin phosphatase activity, thus concluding that gracilins are a “valuable option for synthetic drug development” [212,213]. Kicha and colleagues determined the immunomodulatory effects of the mycosporine-like amino acids (MAA) shinorine and porphyra-334 (**196**, **197**) isolated from the red alga *Porphyra* sp. and reported they both induced NF-κB activity and moderated tryptophan metabolism in human myelomonocytic cell line THP-1, thus recommending “a more detailed risk-benefit assessment” before MAA are used “in daily care products” [214]. Chung and colleagues studied the cembrane-type diterpenoid sinulariolide (**198**), isolated from the Taiwanese cultured soft coral *Sinularia flexibilis* and observed suppression of LPS-phenotypic maturation, cytokine and NO production and co-stimulatory molecule expression of murine bone marrow-derived dendritic cells, as well as signaling pathways, concluding “that sinulariolide may be utilized in the treatment of autoimmune andinflammatory disorders” [215]. Gao and colleagues determined significant effects of several peptides (**199**) isolated from Indian Ocean deep (1654 m) sediment-derived actinomycete *Williamsia* sp. MCCC 1A11233 on both IgE-sensitized RBL-2H3 mast cell histamine and proinflammatory cytokine release, and a murine model of passive cutaneous anaphylaxis, thus observing that “the three compounds have the potential to prevent or treat IgE-sensitized allergic disorders” [216].

In contrast, for the marine compounds (**200**–**205**) listed in Table 2 and depicted in Figure 2, only the immune bioactivity (IC_50_) was reported, but no molecular mechanism of action had been described by the time of publication: A novel dimeric nitrophenyl *trans*-epoxyamide chrysamide C (**200**) produced by a Indian Ocean deep (3386 m) sediment-derived fungus *Penicillium chrysogenum* SCSIO41001 [217]; a new dimeric macrolide xylopyranoside cocosolide (**201**) isolated from the Guamanian marine cyanobacterium *Symploca* sp. [218]; a new N-acyl dopamine glycoside myxillin A (**202**) isolated from the Icelandic hydrothermal vent-derived sponge *Myxilla incrustans* [219]; the known steroidal saponin pectinioside A (**203**) isolated from the starfish *Patiria pectinifera* [220]; a new benzophenone polyketide peniphenone (**204**) obtained from the mangrove *Sonneratia apetala* endophytic fungus *Penicillium* sp. ZJ-SY_2_ [221]; and a known analog of the antimycin-type depsipeptide somalimycin, named USF-19A (**205**) isolated from the South China Sea deep (3536 m) sediment-derived *Streptomyces somaliensis* SCSIO ZH66 [222].

### 3.4. Marine Compounds Affecting the Nervous System

As shown in Table 2 and Figure 2, in 2016–2017, the preclinical nervous system pharmacology of marine compounds (**206**–**225**) reported several mechanisms of action involving potassium (K^+^) channels, nicotinic acetylcholine, calcium (Ca^2+^) and serotonin receptors, as well as in vivo models of antinociception and neuroprotection.

Five marine compounds were shown to bind K^+^ channels (**206**, **207**), calcium channels (**208**), nicotinic acetylcholine receptors (nACHR) (**209**) and serotonin receptor 5-HT_7_ (**218**). Moreels and colleagues electrophysiologically demonstrated that the novel peptide APETx4 (**206**) isolated from the marine sea anemone *Anthopleura elegantissima* inhibited the K^+^ channel K_v_10.1 by keeping it in a closed state and thus, because the K_v_10.1 is overexpressed in a human tumors, APETx4 could become a “scaffold for design and synthesis of more potent and safer anticancer drugs” [223]. Goda and colleagues conducted detailed studies that determined that the marine xanthophyll carotenoid astaxanthin (**207**) reversed the toxicity of the calcium-dependent Maxi-K (BK) K^+^ channel antagonist food mycotoxin penitrem A (PA) “by restoring normal levels of the targeted neurotransmitters” both in Schwann cells CRL-2765 in vitro and in both the nematode *Caenorhabditis elegans* and rat in vivo models thus concluding astaxanthin might be useful “for in vivo prevention of PA-induced brain toxicity [224]. Mendez and colleagues investigated the marine guanidine alkaloid crambrescidin 816 (**208**) produced by the Mediterranean sponge *Crambe crambe* and known to block voltage dependent L-type Ca^2+^ channels, and observed that cytotoxicity was concomitant with an increase of cytosolic Ca^2+^ in a primary culture of cortical neurons by a mechanism that “was mediated by both NMDA and AMPA glutamate receptor subtypes” activation [225]. Jiang and colleagues discovered a novel O-conotoxin GeXXVIIA linear peptide (**209**) from the venom of the South China sea cone snail *Conus generalis* that potently inhibited the human α9α10 nicotinic acetylcholine receptor (nACh), a nAChR receptor expressed in both the nervous system, as well as in other non-neuronal cells, by a non-competitive and voltage-independent mechanism. The data suggest that this new inhibitor “would facilitate unraveling the functions of this nAChR subtype” [226]. Uchimasu and colleagues reported a novel guanidine alkaloid mellpaladine A (**218**) isolated from a Palauan Didemnidae tunicate that modulated mice behavioral profiles after an intracerebroventricular injection with particular selectivity for the serotonin receptor 5-HT_7_, thus, supporting the notion that “marine alkaloids are unique…source of neuroactive compounds” [239].

Two additional studies extended the pharmacology of conopeptides (**210**–**211**). Bernáldez and colleagues discovered a novel 25-mer peptide member of the γ-conotoxin family PiVIIA (**210**) in the venom of the Mexican marine snail *Conus princepts* that increased Ca^2+^ currents in dorsal root ganglion neurons without affecting the Na^+^, K^+^ or acid sensing ionic channel currents, further studies being proposed to “define its potential use as a positive modulator of neuronal activity” [227]. Brust and colleagues identified a previously uncharacterized nine amino acid conorphin-T (**211**) from a *Conus textile* venom peptide library with selectivity for *κ*-opioid receptors (KOR) that led to the development of several novel KOR agonists, which potently inhibited splanchnic colonic nociceptors, thus, becoming promising leads “for the development of irritable bowel syndrome treatments” [228].

One study contributed to nociceptive pharmacology. Logashina and colleagues reported that the novel peptide Ms 9a-1 (**219**), isolated from the venom of the sea anemone *Metridium senile*, produced significant analgesic and anti-inflammatory effects in a murine thermal hyperalgesia model by a mechanism that potentiated the response of transient receptor potential ankyrin-repeat 1 receptor (TRPA1) and was followed by loss of TRPA1-expressing neurons [240].

The neuroprotective activity of marine compounds (**151**, **212**, **214**, **216**, **220**, **223**, **225**) was reported in ten studies. Three studies reported on the neuroprotective pharmacology of the marine carotenoid fucoxanthin (**151**) isolated from several brown algae: *Undaria pinnatifida, Ecklonia stolonifera* and *Sargassum horneri*: Jung and colleagues studied β-site amyloid precursor protein cleaving enzyme 1 (BACE-1), which is strongly correlated with Alzheimer’s disease, and observed that fucoxanthin inhibited BACE1 activity exhibiting mixed-type inhibition in vitro, while molecular docking simulations showed that two fucoxanthin hydroxyl groups interacted with two BACE1 residues (Gly11 and Ala127), mechanistic studies that suggest fucoxanthin “may be a good template for anti-AD drugs” [232]. Lin and colleagues demonstrated that in mice fucoxanthin reversed scopolamine-induced cognitive impairments by inhibiting brain acetylcholinesterase by a non-competitive mechanism, as well as increasing choline acetyltransferase activity and brain-derived neurotrophic factor (BDNF) expression in hippocampus and cortex. The studies appear to anticipate fucoxanthin’s “therapeutic efficacy for the treatment of AD by acting on multiple targets” [233]. Zhang and colleagues discovered that fucoxanthin provided neuroprotection in an experimental murine model of traumatic brain injury (TBI), “a major public health problem” by alleviating TBI-induced secondary brain injury, cerebral edema and lesions, while also demonstrating that it attenuated TBI-induced apoptosis and oxidative stress “at least partly” via regulation of the nuclear factor erythroid 2-related factor (Nrf2)-antioxidant-response element (ARE) and Nrf2-autophagy pathways, thus, making fucoxanthin “an attractive therapeutic agent in the treatment of TBI in the future” [234]. Two studies were reported by the same research group describing the neuroprotective and anti-inflammatory effects of the cembranolide analog 11-dehydrosinulariolide (11-de) (**212**) isolated from the soft coral *Sinularia flexibilis*: Feng and colleagues observed that 11-de increased the expression of BDNF in a neuroblastoma cell line in vitro and had a protective effect in both an in vivo zebrafish and rat Parkinson’s disease model, results that the investigators hoped would “help treat patients diagnosed with Parkinson’s disease” [229]. Chen and colleagues reported that 11-de improved the functional recovery in a rat thoracic spinal cord contusion injury experimental model with an antiapoptotic and anti-inflammatory mechanism that attenuated iNOS and TNF-α, thus, proposing “this compound may be a promising therapeutic agent for spinal cord injury” [230]. Fang and colleagues showed that a new cyclopentenone 5-hydroxycyclopenicillone (**214**) isolated from a culture of the marine sponge *Hymeniacidon perleve*-derived fungus *Trichoderma* sp. HPQJ-34 acted as a moderate free radical scavenger, and had anti-Aβ fibrillization and neuroprotective properties, concluding that this compound “might be of interest to neuropharmacology research and anti-AD drug discovery programs” [235]. Alonso and colleagues evaluated the pyrroloiminoquinone makaluvamine J (**216**), isolated from a Fijian marine sponge *Zyzzya* sp. and showed that it “provided full neuroprotection”, as it potently reduced mitochondrial damage by reactive oxygen species, as well as improved endogenous glutathione and catalase in mouse and human neuronal models, thus, potentially contributing to “antioxidant therapies in neurodegenerative diseases” [237]. Kim and colleagues found that the phlorotannin phlorofucofuroeckol (**220**) isolated from the brown seaweed *Ecklonia cava*, increased cell viability in glutamate-treated rat adrenal phaeochromocytoma PC12 cells by inhibiting apoptotic cell death as well as mitochondrial reactive oxygen species generation, results, which taken together, suggest PFF may be developed as “ a neuroprotective agent in ischemic stroke” [241]. Perni and colleagues discovered that the aminosterol squalamine (**223**) isolated from the dogfish shark *Squalus acanthias* displaced the “intrinsically disordered” protein α-synuclein, associated with Parkinsons’s disease, from lipid vesicles and membranes by competitively binding at the surfaces, as well as suppressed muscle paralysis in a nematode worm *Caenorhabditis elegans* strain oversexpressing α-synuclein, suggesting squalamine “could be a means for a therapeutic intervention in Parkinsons’s disease” [244]. Pan and colleagues extended the pharmacology of xyloketal B (**225**) isolated from the mangrove fungus *Xylaria* sp. by investigating effects of xyloketal in an adult mice stroke model, observing that pre- and post-treatment treatment reduced both brain infarct volume and the generation of ROS and pro-inflammatory cytokines by suppression of ROS/TLR4/NF-κB inflammatory signaling pathway, thus providing evidence for “potential application of xyloketal B in stroke therapy” [246].

As shown in Table 2, five marine compounds were shown to modulate *other* molecular targets, i.e., the acetylcholinesterases (**213**, **217**, **224**), endogenous transient receptor potential canonical type 1 channel (TRPC1) (**215**), human monoamine oxidase B enzyme (**221**), and prolonged synaptic transmission (**222**). Botic Lee and colleagues discovered that the brominated alkaloid discorhabdin G (**213**) isolated from the Antarctic *Latrunculia biformis* sponge inhibited electric eel and human acetylcholinesterases by a reversible and competitive mechanism, observations that could potentially lead to new Alzheimer’s disease “cholinesterase inhibitors based on the scaffold of discorhabdins” [231]. Adelhameed and colleagues reported the isolation and structure elucidation of a new phytoceramide MEC-1-4 (**217**) from the Egyptian Red Sea sponge *Mycale euplectellioides* that moderately inhibited acetylcholinesterase by interacting with the enzyme “via hydrogen bonding, hydrophobic contacts and hydrophilic-hydrophobic interactions”, thus suggesting MEC-1-4 might become a “valuable lead compound for AD management” [238]. Moodie and colleagues evaluated the known brominated phenethylamine derivative stryphnusin (**224**) isolated from the Norwegian sponge *Stryphus fortis* that moderately inhibited electric eel acetylcholinesterase by a reversible competitive mechanism and with no effect on muscle function or neuromuscular transmission thus contributing to novel and promising approaches for “symptomatic treatment of AD” [245]. Flores and colleagues investigated the marine polyether toxin maitotoxin (**215**) produced by the dinoflagellate *Gambierdiscus toxicus* that is responsible for ciguatera fish poisoning, demonstrating that its mechanism of action involves activation of the non-voltage-gated cation TRPC1 in *X. laevis* oocytes and thus proposing this toxin as a “useful tool for further studies of TRPC1 channels” [236]. Lee and colleagues reported that the polyketide piloquinone (**221**) isolated from a Californian marine sediment-derived *Streptomyces* sp. CNQ-027 potently inhibited recombinant human monoamine oxidase B enzyme considered a “target in AD and Parkinson’s disease”, by a competitive and reversible mechanism which highlighted the importance of the “ester functionality in the ring system for bioactivity”, thus possibly becoming “a new potential lead compound for the development of MAO inhibitors” [242]. Caplan and colleagues extended the pharmacology of the marine diterpene glycoside pseudopterosin A (**222**) isolated from the Bahamanian gorgonian soft coral *Pseudopterogorgia elisabethae* by demonstrating it prolonged synaptic transmission in an experimental oxidative stress model, as well as extensively distributed in murine brain, findings that suggested a “potential as a novel neuromodulatory agent” [243].

Finally, and as shown in Table 2, several marine compounds (**226**–**234**) affected the nervous system, but their detailed molecular mechanisms of action remained undetermined at the time of publication: a novel Indonesian *A**plysinellidae* sponge-derived bromotyrosine araplysillin X (**226**) that inhibited the aspartic protease BACE1 involved in AD [247]; a novel Panamanian marine cyanobacterium cf. *Symploca* sp.-derived terpene caracolamide A (**227**) with in vitro calcium influx and calcium channel oscillation modulatory activity [248]; a Mexican marine cone snail *Conus spurius* peptide conorfamide-Sr3 (**228**) shown to block *Shaker* subtype voltage-gated potassium channels [249]; a new conopeptide contryphan-Bt (**229**) isolated from the South China Sea cone snail *Conus betulinus* shown to be neurotoxic to mice [250]; acetylcholinesterase inhibitory activity in two know meroterpenoids, dehydroaustin (**230**) isolated from the Chinese mangrove endophytic fungus *Aspergillus* sp. 16-5c [251], and terreulactone C (**233**) isolated from the South China Sea mangrove-derived endophytic fungus *Penicillium* sp. SK5GW1L [254]; a known alkaloid hymenidin (**231**), originally isolated from the Caribbean marine sponge *Agelas citrina* that inhibited voltage-gated potassium channels [252]; a neuroprotective and known bromotyrosine alkaloid psammaplysene A (**232**) originally isolated from the marine sponge *Psammaplysilla* sp. [253]; and inhibition of α9α10 nicotinic acetylcholine receptor by turripeptide (**234**) isolated from the Philippine marine gastropod *Unedogemmula bisaya* venom [255].

## 4. Marine Compounds with Miscellaneous Mechanisms of Action

Further 2016–2017 preclinical pharmacology for marine compounds (**49**, **51)** as well as that of 79 compounds (**235**–**313**) with miscellaneous mechanisms of action is shown in Table 3, with their corresponding structures, presented in Figure 3. Given that, at the time of publication, a comprehensive pharmacological characterization of these compounds remained unavailable, their assignment to a particular drug class will probably require further investigation into their molecular mechanism of action.

Table 3 presents not only the pharmacological activity (an IC_50_), but also the molecular mechanism of action of the following marine natural compounds: Fungus *Acremonium* sp. (F9A015) polyketide acredinone C (**235**) [256]; algal terpenoid astaxanthin (**207**) [257]; sponge *Axinyssa* sp. bisabolene sesquiterpene (**236**) [258]; sponge *Carteriospongia* sp. scalarane sesterterpenoid (**237**) [259]; fungus *Aspergillus ungui* NKH-007 depsidone 7-chlorofolipastatin (**238**) [260]; sponge *Acanthostrongylophora ingens* halogenated alkaloid chloromethylhalicyclamine B (**239**) [261]; cyanobacterium *Leptolyngbya* sp. depsipeptide coibamide A (**240**) [262]; sponge *Theonella* aff. *swinhoei* cyclic peptide cyclotheonellazole A (**241**) [263]; bacterium SNA-024 *N^6^*,*N^6^*-dimethyladenosine (**242**) [264]; gorgonian *Briareum excavatum* briarane-type diterpene excavatolide B (**150**) [265]; brown algae *Undaria pinnatifida, Ecklonia stolonifera* and/or *Sargassum horneri* marine carotenoid fucoxanthin (**51**) [266]; sponge *Phorbas* sp. diterpenoid gagunin D (**243**) [267]; bacterium *Bacillus* sp. strain SCO-147 (-)-4-hydroxysattabacin (**244**) [268]; cyanobacteria *Lyngbya majuscula* and *Tolypothrix* sp. γ-pyrone kalkipyrone (**245**) [269]; sponge *Stylissacarteri* sp. alkaloid latonduine A (**246**) [270]; sponge *Negombata* sp. alkaloid latrunculin A (**247**) [271]; bacterium *Streptomyces* sp. polyketide nahuoic acid A (**248**) [272]; bacterium *Streptomyces* sp. YP127 napyradiomycin A1 (**49**) [273]; sponge *Leucetta microraphis* alkaloid leucettamine B-related synthetic analogue polyandrocarpamine A (**249**) [274]; sponge *Cacospongia* sp. terpenoid scalaradial (**250**) [275]; fungus *Stachybotrys* sp. KCB13F013 meroterpenoid stachybotrysin (**251**) [276]; soft coral *Clavularia* sp. terpenoid stolonidiol (**252**) [277]; cyanobacterium *Lyngbya* sp. peptide tasiamide F (**253**) [278]; sponge *Theonella* sp. cyclic peptide theonellamide A (**254**) [279]; fungus *Penicillium* sp. HL-85-ALS5-R004 polyketide toluquinol (**255**) [280]; fungus *Penicillium janthinellum* alkaloid *N*-Me-trichodermamide B (**256**) [281]; gorgonian-derived fungus *Aspergillus versicolor* cyclopeptides versicotides D-F (**257**–**259**) [282]; sponges *Ircinia* and *Spongia* spp. furanoterpene (7*E*, 12*E*, 20*Z*, 18*S*)-variabilin (**260**) [283]; and fungus *Xylaria* sp. terpenoid xyloketal B (**225**) [284].

Also presented in Table 3 is the pharmacological activity (IC_50_ for enzyme or receptor inhibition) of marine-derived compounds (**261**–**313**), although their respective mechanisms of action remained undetermined and will require further investigation: sponge *Aaptos aaptos* alkaloid 9-methoxyaaptamine (**261**) [285]; mangrove-derived fungus *Ascomycota* sp. SK2YWS-L polyketide ascomindone A (**262**) [286]; mudflat-derived fungus *Aspergillus niger* polyketide aurasperone B (**263**) [287]; alga-derived fungi *Penicillium thomii* and *Penicillium lividum* meroterpenoid austalide H acid ethyl ester (**264**) [288]; fungus *Biscogniauxia mediterranea* peptide biscogniauxone (**265**) [289]; fungus *Aspergillus ustus* sesterpenoid cerebroside D (**266**) [290]; sponge *Spongia ceylonensis* diterpene ceylonin A (**267**) [291]; sponge-derived fungus *Stachybotrys chartarum* sesquiterpene chartarene D (**268**) [292]; sponge-associated fungus *Talaromyces stipitatus* KUFA 0207 anthraquinone citreorosein (**269**) [293]; fungus *Aspergillus* sp. SCSIOW3 polyketide cordyol (**270**) [294]; fungus *Chaetomium cristatum* dioxopiperazine alkaloid cristazine (**271**) [295]; sponge-associated fungus *Emericella variecolor* polyketides diasteltoxins A-C (**272**–**274**) [296]; sponge *Latrunculia* sp. pyrroloiminoquinone alkaloid discorhabdin L (**275**) [297]; sponge *Dysidea* sp. polybrominated diphenyl ether 3,4,5-tribromo-2-(2′,4′-dibromophenoxy)-phenol (**276**) [298]; sponge *Dysidea* sp. tetracyclic meroterpene dysiherbol A (**277**) [299]; sand dollar *Scaphechinus mirabilis* aminonaphthoquinone echinamine B (**278**) [300]; sponge derived fungus *Stachylidium* sp. tetrapeptide endolide B (**279**) [301]; mangrove-derived fungus *Eurotium rubrum* MA-150 (±)-europhenol A (**280**) [302]; sponge *Fascaplysinopsis* sp. bis-indole alkaloid fascaplysin (**281**) [303]; nudibranch *Jorunna funebris* tetrahydroisoquinolinequinone alkaloid fennebricin A (**282**) [304]; sponge *Petrosia* sp. depsipeptide halicylindramide A (**283**) [305]; sponge *Halichondria* cf. *panicea* polyacetylene isopetrosynol (**284**) [306]; cyanobacterium *Leptolyngbya* sp. macrolide polyketide leptolyngbyolide B (**285**) [307]; ascidian *Lissoclinum mandelai* macrocyclic polyketide mandelalide C (**286**) [308]; sponge *Hyrtios digitatus* tetracyclic merosesquiterpene 19-methoxy-9,15-ene-puupehenol (**287**) [309]; sponge *Monanchora pulchra* cyclic guanidine alkaloid monanchomycalin B (**288**) [310]; sponge *Plakortis simplex* polyketide monotriajaponide A (**289**) [311]; sponge *Mycale lissochela* terpenoid mycalenitrile-15 (**290**) [312]; sponge *Hyrtios* sp. meroterpenoid nakijinol G (**291**) [313]; sponge *Theonella swinhoei* cyclic pentapeptide nazumazole D (**292**) [314]; fungus *Aspergillus* sp. LF660 benzocoumarin polyketide pannorin (**293**) [315]; fungus *Alternaria* sp. NH-F6 polyketides perylenequinones (**294**, **295**) [316]; sponge *Petrosia alfiani* xestoquinone derivatives petroquinones A and B (**296**, **297**) [317]; fungus *Peyronellaea glomerata* isocoumarin derivatives peyroisocoumarins B and D (**298**, **299**) [318]; fungus *Phoma* sp. NTOU4195 polyketide phomaketide A (**300**) [319]; ascidian *Sidnyum elegans* phosphorylated polyketide phosphoeleganin (**301**) [320]; soft coral *Pseudopterogorgia rigida* sesquiterpenes (**302**, **303**) [321]; sponge *Spongia ceylonensis* diterpene *ent*-13-norisocopalen-15-al-18-oic acid (**304**) [322]; bryozoan *Schizomavella mamillata* 5-alkylresorcinol derivatives schizols A and B (**305**, **306**) [323]; fungus *Stachybotrys longispora* FG216 isoindolinone derivatives FGFC6 and FGFC7 (**307**, **308**) [324]; sponge *Spongia pertusa* Esper sesquiterpene (**309**) [325]; sponge *Psammocinia* sp. furanosesterterpene tetronic acid sulawesin A (**310**) [326]; cyanobacterium *Okeania* sp. cyclic depsipeptide urumamide (**311**) [327]; gorgonian-derived fungus *Aspergillus versicolor* LZD-14-1 alkaloid versiquinazoline B (**312**) [328]; and sponge *Xestospongia testudinaria* new bioactive steroidal ketone (**313**) [329].


marinedrugs-19-00049-t003_Table 3Table 3Marine pharmacology in 2016–2017: marine compounds with miscellaneous mechanisms of action.Compound/Organism ^a^ChemistryPharmacological ActivityIC_50_
^b^MMOA ^c^Country ^d^Referencesacredinone C (**235**)/fungusPolyketide ^e^Osteoclast differentiation induction inhibition10 μM *NFATc1 transcription inhibitionS. KOR[256]astaxanthin (**207**)/shrimpTerpenoid ^f^Hepatic stellate cell activation inhibition10 μM *Decreased ROS and NOX2 expression reductionUSA[257]*Axinyssa* sp. bisabolene (**236**)/spongeTerpenoid ^f^PTP1B inhibition1.9 μMAkt phosphorylationJPN[258]*Carteriospongia* sp. terpenoid (**237**)/spongeTerpenoid ^f^Apoptosis induction0.06 μg/mL *Topoisomerase IIα and Hsp90 inhibitionEGY, SWE, TWN[259]7-chlorofolipastatin (**238**)/fungusPolyketide ^e^Macrophage SOAT 1 inhibition6.8 μMSOAT 1 and 2 inhibition *in vitro*JPN[260]chloromethylhalicyclamine B (**239**)/spongeAlkaloid ^g^Protein kinase CK1δ/ε inhibition6 μMATP-binding site dockingFRA, ITA, NLD[261]coibamide A (**240**)/cyanobacteriumPeptide ^f^VEGFA secretion inhibition<5 nMAntiangiogenic propertiesUSA[262]cyclotheonellazole A (**241**)/spongePeptide ^f^chymotrypsin and elastase inhibition0.034–0.62 nMEnzyme S2 subsite bindingBEL, ISR, NLD[263]*N*^6^,*N*^6^-dimethyladenosine (**242**)/bacteriumAlkaloid ^g^AKT phosphorylation inhibition5 μM *S473 site inhibitionUSA[264]excavatolide B (**150**)/soft coralTerpenoid ^f^Modulation of atrial myocytes10 μM *Ca^2+^ homeostasis modulationTWN[265]fucoxanthin (**51**)/algaTerpenoid ^f^Lung fibrosis attenuation10 mg/kg ***Type 1 collagen expression decreaseS. KOR[266]gagunin D (**243**)/spongeTerpenoid ^f^Melanin synthesis inhibition12.7 µMTyrosinase expression inhibitionS. KOR[267](−)-4-hydroxysattabacin (**244**)/bacteriumPolyketide ^e^Melanin synthesis inhibition25 μg/mL *Tyrosinase, TRP-1 and TRP-2 expression inhibitionS. KOR[268]kalkipyrone (**245**)/cyanobacteriumPolyketide ^e^Adipose tissue suppression5 mg/kg ***Enhance LA plasma levelsJPN[269]latonduine A (**246**)/spongeAlkaloid ^g^CTFR inhibition62 nMPARP isozymes inhibitionCAN, GBR[270]latrunculin A (**247**)/spongeAlkaloid ^g^ECFC tube inhibition0.043 µMSpecific kinases inhibitionUSA[271]nahuoic acid (**248**)/bacteriumPolyketide ^e^SETD8 inhibition6.5 µMCompetitive inhibition of SAM bindingCAN, PNG[272]napyradiomycin A1 (**49**)/bacteriumTerpenoid ^f^Angiogenesis inhibition10 µMVE-cadherin inhibitionS. KOR[273]polyandrocarpamine A (**249**)/spongeAlkaloid ^g^DYRK and CLK selective inhibition0.17–0.93 µMCyclin D1 phosphorylation inhibitionAUS, BRA, DEU, FRA[274]scalaradial (**250**)/spongeTerpenoid ^f^TRPM2 ion channel inhibition0.2 µMLack of PLA_2_ inhibitionJPN, NZL, USA[275]stachybotrysin (**251**)/fungusTerpenoid ^f^Osteoclast differentiation inhibition5 μg/mL *MAPK kinase pathway inhibitionJPN, S. KOR[276]stolonidiol (**252**)/soft coralTerpenoid ^f^PKCα membrane translocation5 µM *Increased ChAT activityUSA[277]tasiamide F (**253**)/cyanobacteriumPeptide ^g^Cathepsin D and E inhibition23–57 nMDocking studies completedUSA[278]theonellamide A (**254**)/spongePeptide ^g^Bilipid membrane disruption20 µM *Binding to sterolsJPN[279]toluquinol (**255**)/fungusPolyketide ^e^Lymphangiogenesis inhibition6.2 µMSuppression of Akt and ERK ½ phosphorylationBEL[280]*N*-Me-trichodermamide B (**256**)/fungusAlkaloid ^g^H_2_O_2_ oxidative damage inhibition5 µM *Nrf2-signaling regulationCHN[281]versicotides D–F (**257**–**259**)/fungusPeptide ^g^Foam cell formation inhibition10 µM *Cholesterol influx inhibitionCHN[282]variabilin (**260**)/spongeTerpenoid ^f^PTP1B inhibition1.5 μMTCPTP inhibitionIND, JPN[283]xyloketal B (**225**)/fungusTerpenoid ^f^NAFLD attenuation5 mg/kg ***SREBP-1c expression inhibitionCHN[284]9-methoxyaaptamine (**261**)/spongeAlkaloid ^g^PPRE activation0.039 μg/mL *UndeterminedIDN, MYS[285]ascomindone A (**262**)/fungusPolyketide ^e^DPPH radical scavenging inhibition18.1 μMUndeterminedCHN[286]aurasperone B (**263**)/fungusPolyketide ^e^DPPH radical scavenging inhibition0.01 μMUndeterminedS. KOR[287]austalide H acid ethy ester (**264**)/ fungusTerpenoid ^f^*Endo*-1,3-β-_D_-glucanase inhibition0.2 μMUndeterminedRUS[288]*B. mediterranea* cyclopentapeptide (**265**)/fungusPeptide ^g^GSK-3β inhibition8.04 μMUndeterminedCHN, DEU[289]cerebroside (**266**)/fungusPolyketide ^e^Spermatozoa inhibition8 μMUndeterminedRUS[290]ceylonin A (**267**)/spongeTerpenoid ^f^Osteoclast inhibition<50 μM *UndeterminedNLD, JPN
chartarene D (**268**)/fungusTerpenoid ^f^Tyrosine kinases inhibition0.1–0.8 μMUndeterminedCHN, DEU[292]citreorosein (**269**)/fungusPolyketide ^e^Anti-obesity activity0.17 μMUndeterminedGBR, PRT, THAI[293]cordyol C (**270**)/fungusPolyketide ^e^Erythrocyte biomembrane protection4.9 μMUndeterminedCHN[294]cristazine (**271**)/fungusAlkaloid ^g^DHHP radical scavenging19 μMUndeterminedS. KOR[295]diasteltoxins A–C (**272**–**274**)/fungusPolyketide ^e^Thioredoxin reductase inhibition7.2–12.8 μMUndeterminedCHN, DEU[296]discorhabdin L (**275**)/spongeAlkaloid ^g^HIF-1α transcription inhibition0.73 μMUndeterminedNZL, USA[297]*Dysidea* sp. diphenyl ether (**276**)/spongePolyketide ^e^Mitochondrial complex II inhibition6.4 nMUndeterminedJPN[298]dysiherbol A (**277**)/spongeTerpenoid ^f^NF-κB inhibition0.49 μMUndeterminedAUS, CHN[299]echinamine B (**278**)/sea urchinPolyketide ^e^DHHP radical scavenging6.5 μMUndeterminedRUS[300]endolide B (**279**)/fungusPeptide ^g^Serotonin receptor 5HT_2b_ inhibition0.77 μM **UndeterminedDEU[301]europhenol A (**280**)/fungusPolyketide ^e^DHHP radical scavenging1.23 μg/mLUndeterminedCHN, HUN[302]fascaplysin (**281**)/spongeAlkaloid ^g^P-glycoprotein induction1 μM *UndeterminedIDN[303]fennebricin A (**282**)/nudibranchAlkaloid ^g^NF-κB inhibition1 μMUndeterminedCHN, HUN[304]halicylindramide A(**283**)/spongePeptide ^g^FXR receptor inhibition0.5 μMUndeterminedAUS, S.KOR[305]isopetrosynol (**284**)/spongePolyketide ^e^PTP1B inhibition8.2 μMUndeterminedIDN, JPN[306]leptolyngbyolide B (**285**)/cyanobacteriumPolyketide ^e^F-actin depolymerization11.6 μMUndeterminedJPN[307]mandelalide C (**286**)/ascidianPolyketide ^e^Mitochondrial complex V inhibition3.4 μMUndeterminedUSA[308]19-methoxy-9,15-ene-puupehenol (**287**)/spongeTerpenoid ^f^SR-B1 receptor activation1.78 μMUndeterminedAUS, MYS[309]monanchomycalin B (**288**)/spongeAlkaloid ^g^TRPV1, 2 and 3 receptor inhibition2.8–6.0 μMUndeterminedRUS, S. KOR[310]monotriajaponide A (**289**)/spongePolyketide ^e^PPAR-α and -β activation12.5 μM *UndeterminedCHN, ITA, USA[311]mycalenitrile-15 (**290**)/spongeTerpenoid ^f^PTP1B inhibition8.6 μM *UndeterminedCHN, ITA[312]nakijinol G (**291**)/spongeTerpenoid ^f^PTP1B inhibition4.8 μMUndeterminedCHN[313]nazumazole D (**292**)/spongePeptide ^g^chymotrypsin activity inhibition2 μMUndeterminedJPN[314]pannorin (**293**)/fungusPolyketide ^e^GSK-3β inhibition0.35 µMUndeterminedDEU[315]perylenequinones (**294**, **295**)/fungusPolyketide ^e^BRD4 protein inhibition10 μM *UndeterminedCHN[316]petroquinones A and B (**296**, **297**)/spongePolyketide ^e^USP7 inhibition0.13–2.0 μMUndeterminedIND, JPN, NLD[317]peyroisocoumarins B and D (**298**, **299**)/fungusPolyketide ^e^ARE expression induction10 μM *UndeterminedCHN, DEU[318]phomaketide A (**300**)/fungusPolyketide ^e^Angiogenesis inhibition8.1 μMUndeterminedTWN[319]phosphoeleganin (**301**)/ascidianPolyketide ^e^PTP1B inhibition11 μMUndeterminedCHN, ITA[320]*P. rigida* sesquiterpenes (**302**, **303**)/soft coralTerpenoid ^f^CDC25 phosphatases12–3.4 μMUndeterminedGRC, FRA[321]*S. ceylonensis* diterpene (**304**)/spongeTerpenoid ^f^USP7 inhibition8.2 µMUndeterminedEGY, JPN[322]schizols A and B (**305**, **306**)/bryozoaPolyketide ^e^ABTS cation radical inhibition6.2–7.6 µMUndeterminedESP[323]*S. longispora* isoindolinones (**307**, **308**)/fungusAlkaloid ^g^Fibrinolytic activity25 μg/mL *UndeterminedCHN[324]*S. pertusa* quinone (**309**)/spongeTerpenoid ^f^CDK-2 inhibition4.8 µM **UndeterminedCHN[325]sulawesin A (**310**)/spongeTerpenoid ^f^USP7 inhibition2.8 μMUndeterminedEGY, IDN, JPN, NLD[326]urumamide (**311**)/cyanobacteriumPeptide ^g^Chymotrypsin inhibition33 μMUndeterminedJPN[327]versiquinazoline B (**312**)/fungusAlkaloid ^g^Thioredoxin reductase inhibition12 µMUndeterminedCHN, DEU[328]*X. testudinaria* steroidal ketone (**313**)/ spongeTerpenoid ^f^PTP1B inhibition4.27 μMUndeterminedCHN[329]**^a^ Organism**, *Kingdom Animalia*: shrimp (Phylum Arthropoda); bryozoan (Phylum Bryozoa); ascidian (Phylum Chordata), hydroids, soft corals (Phylum Cnidaria), sea urchin (Phylum Echinodermata), sponge (Phylum Porifera); *Kingdom Fungi*: fungus; *Kingdom Plantae:* alga; *Kingdom Monera*: bacterium; **^b^ IC_50_**: concentration of a compound required for 50% inhibition in vitro; *: estimated IC_50_; ** *K*_d_: equilibrium constant ratio of forward and reverse constants; *K*_i_: concentration needed to reduce the activity of an enzyme by half.*** in vivo study; **^c^ MMOA**: molecular mechanism of action; **^d^ Country:** AUS: Australia; BEL: Belgium; BRA: Brazil; CAN: Canada; CHN: China; DEU: Germany; EGY: Egypt; FRA: France; ESP: Spain; GBR: United Kingdom; GRC: Greece; HUN: Hungary; IDN: Indonesia; ISR: Israel; ITA: Italy; JPN: Japan; MYS: Malaysia; NLD: The Netherlands; NZL: New Zealand; PNG: Papua New Guinea; PRT: Portugal; RUS: Russian Federation; S. KOR: South Korea; SWE: Sweden; THAI: Thailand; TWN: Taiwan; **Chemistry: ^e^** Polyketide; **^f^** Terpene; **^g^** Nitrogen-containing compound; **Abbreviations:** ABTS: 2,2′-azinobis(3-ethylbenzothiazoline-6-sulphonic acid); Akt: protein kinase B; ARE: antioxidant response element; BRD4: bromodomain-containing protein 4; CDK: cyclin-dependent kinase; ChAT: choline acetyltansferase; CTFR: cystic fibrosis transmembrane conductance regulator; CLK: cdc2-like kinases; DDYRK: dual-specificity, tyrosine phosphorylation regulated kinase; DPPH: α, α-diphenyl-β-picrylhydrazyl; ECFC: endothelial colony-forming cell; ERK: extracellular signal-regulated kinase; FXR: farnesoid X receptor; GSK-3β: glycogen synthase kinase 3; Hsp90: heat shock protein 90; MAPK: mitogen-activated protein kinase; NAFLD: nonalcoholic fatty liver disease; NFATc1: nuclear factor of activated T cells, cytoplasmic 1; NF-κB: nuclear factor kappa-light-chain-enhancer of activated B cells; NO: nitric oxide; NOX2: NADPH oxidase 2; Nrf2:nuclear factor-erythroid 2-related factor 2; PARP: poly-ADP ribose polymerase; PKC: protein kinase C; PPAR: peroxisome proliferator-activated receptor; PTP1B: protein tyrosine phosphatase 1B; PPRE: peroxisome proliferator activated receptor response element; ROS: reactive oxygen species; SERCA2A: SR Ca^2+^ ATPase 2A; SETD8: lysine histone methyltransferase 5A; SOAT: sterol *O*-acyltransferase; SREBP-1c: sterol regulatory element-binding protein-1c;TLR5: Toll-like receptor 5; TCPTP: T-cell protein tyrosine phosphatase;TRP: tyrosinase-related protein; TRPM2: melastatin-like transient receptor potential ion channel; USP7: ubiquitin-specific protease 7; VEGF: vascular endothelial growth factor.



Figure 3Marine pharmacology in 2016–2017: Marine compounds with miscellaneous mechanisms of action.
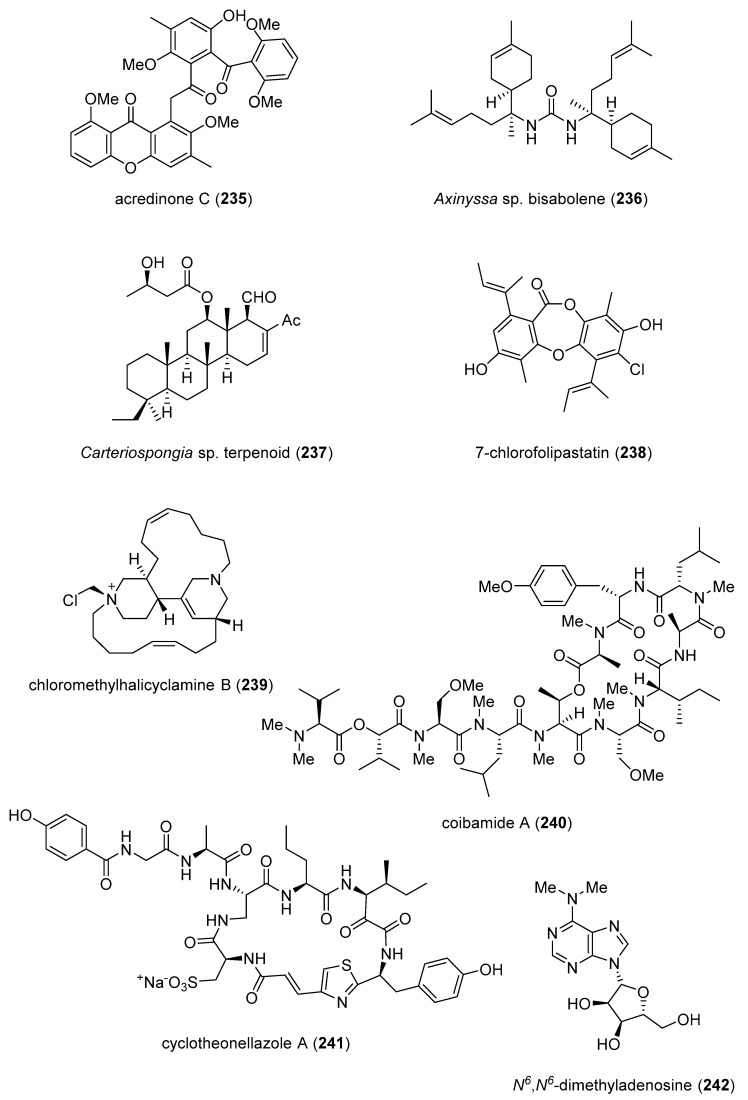

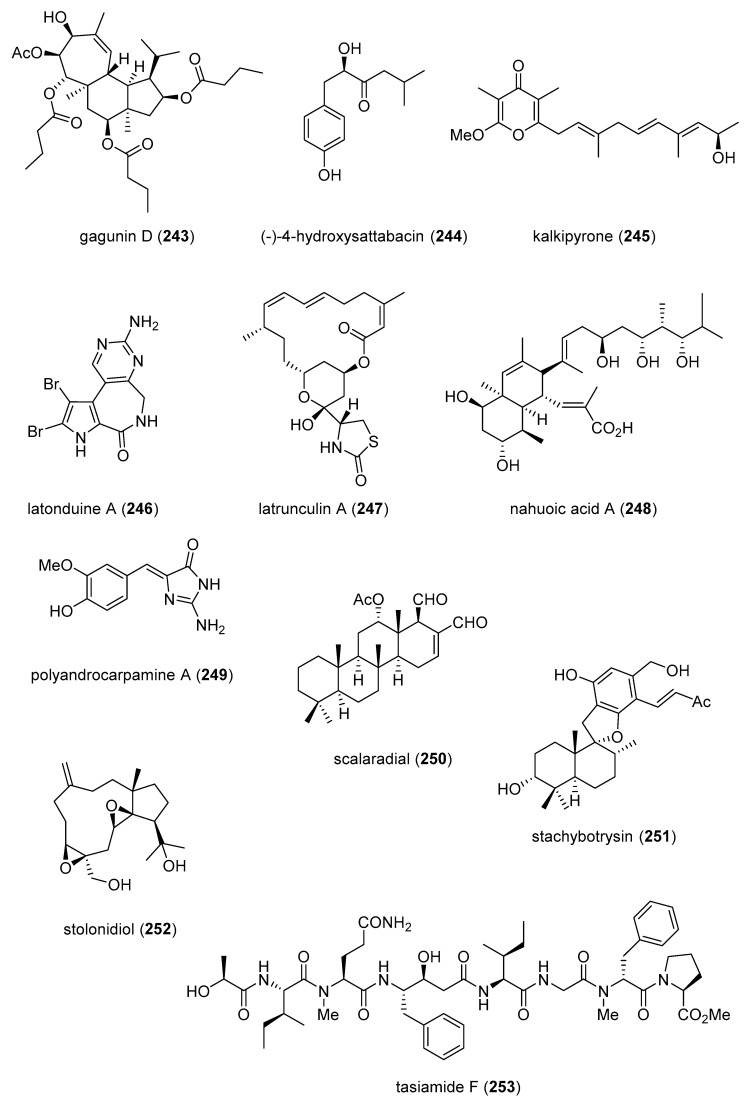

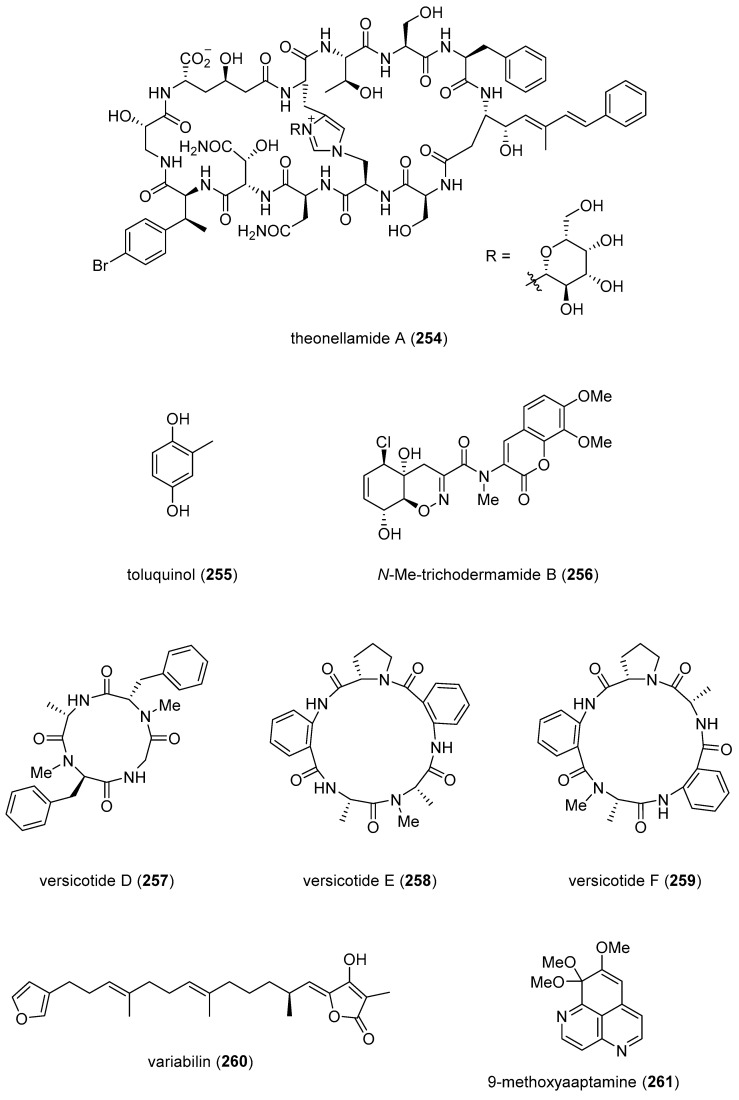

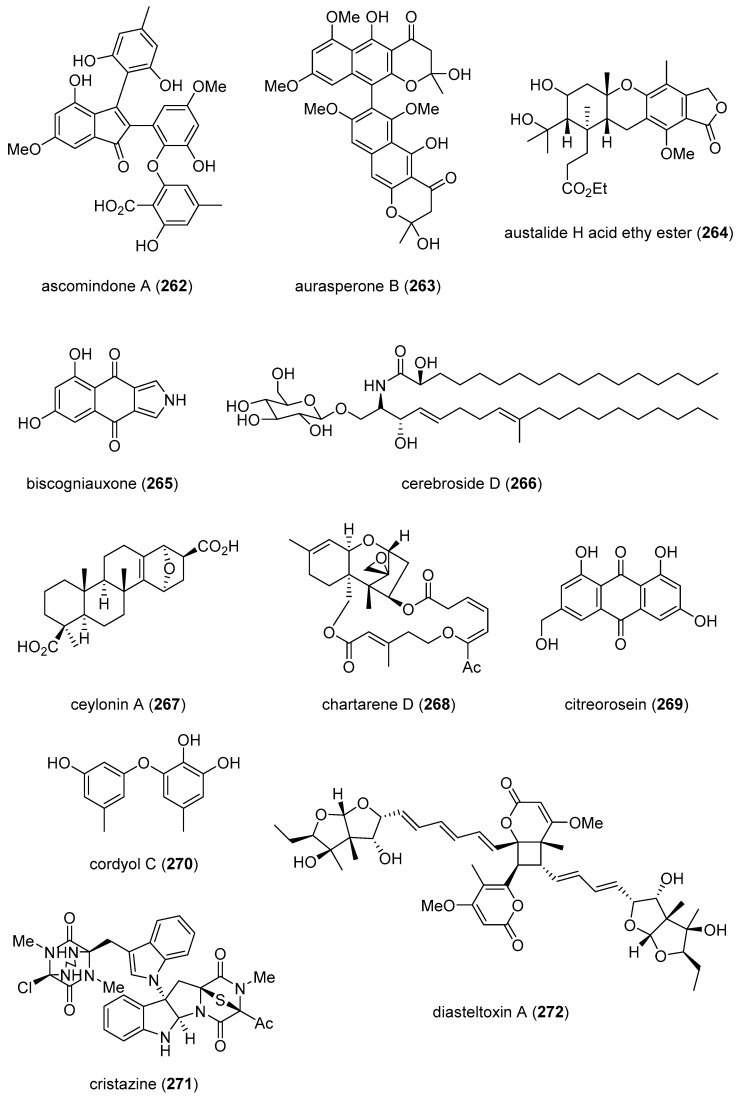

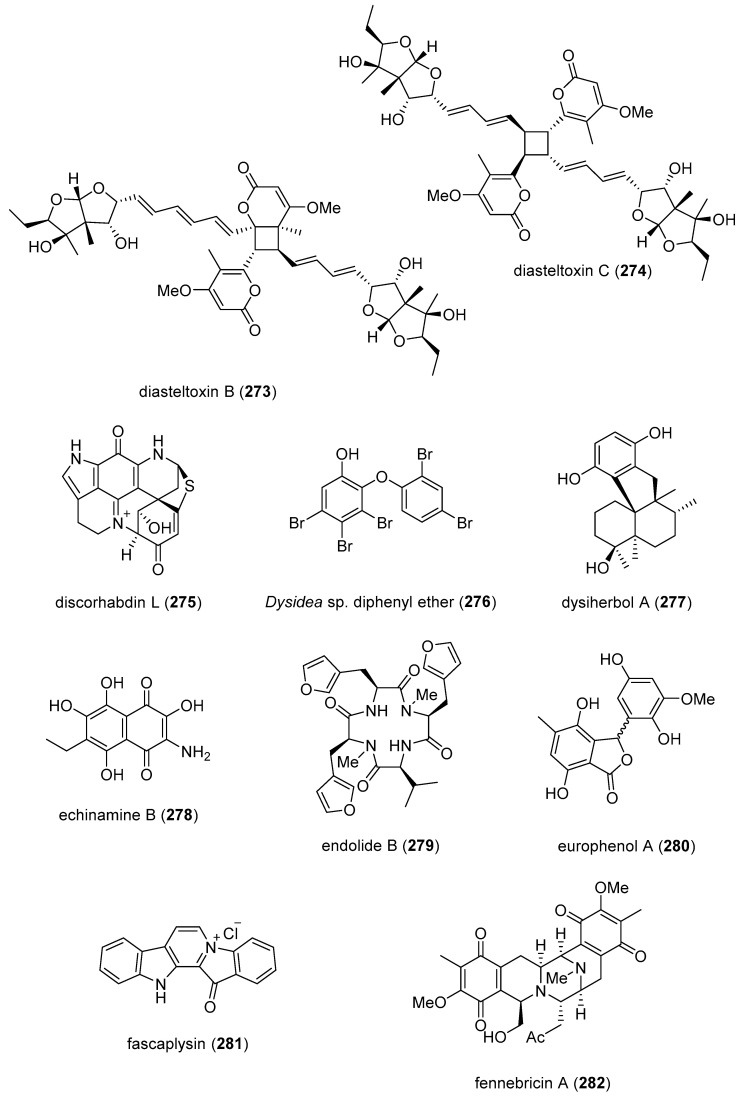

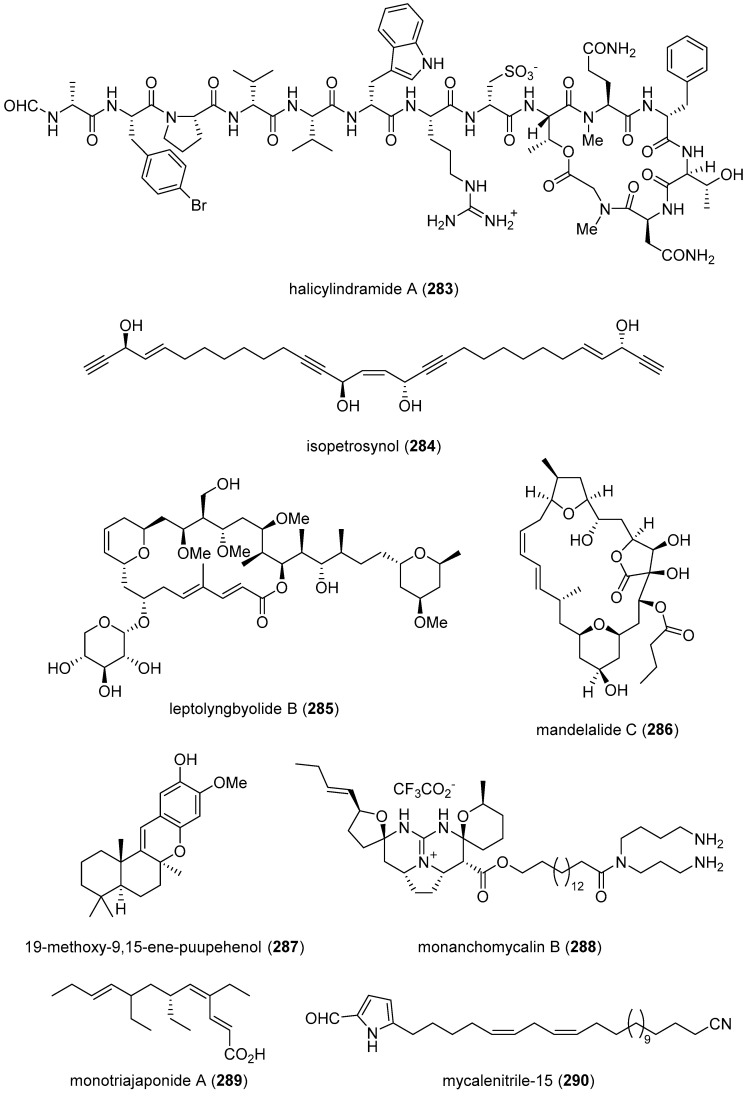

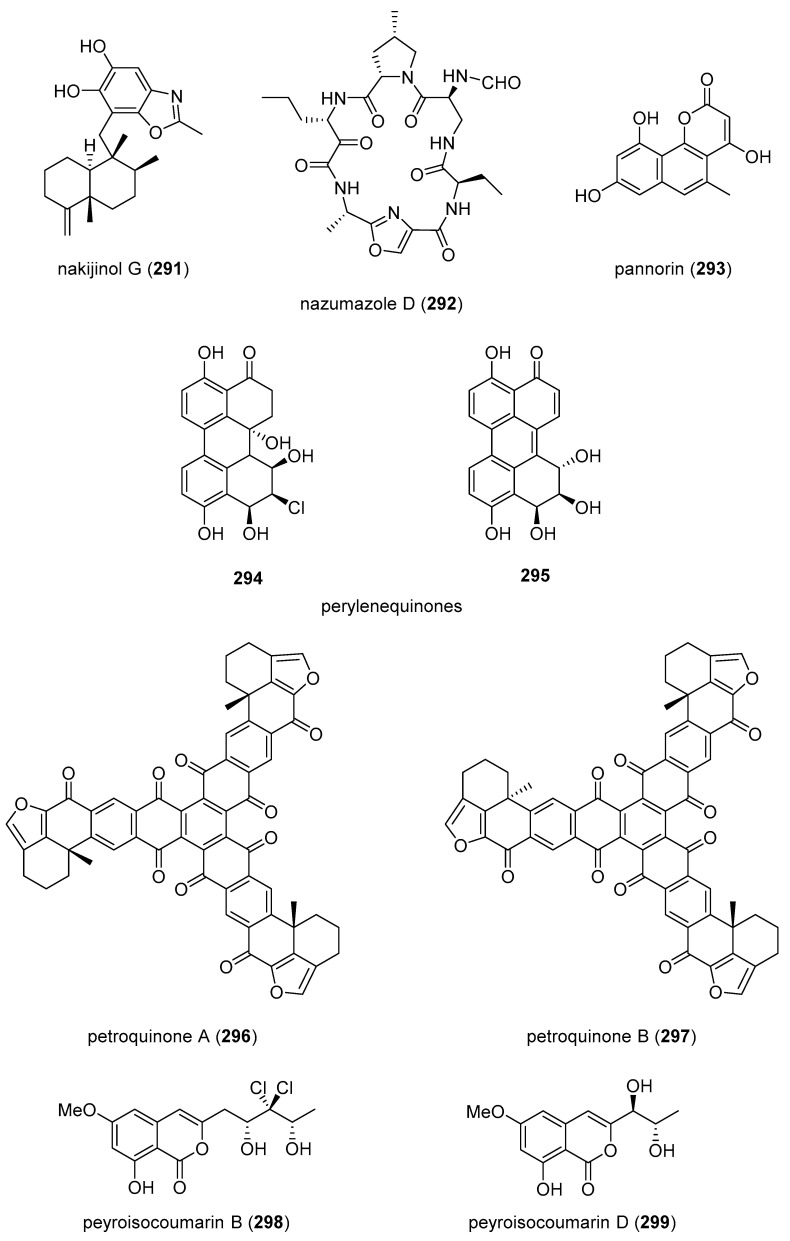

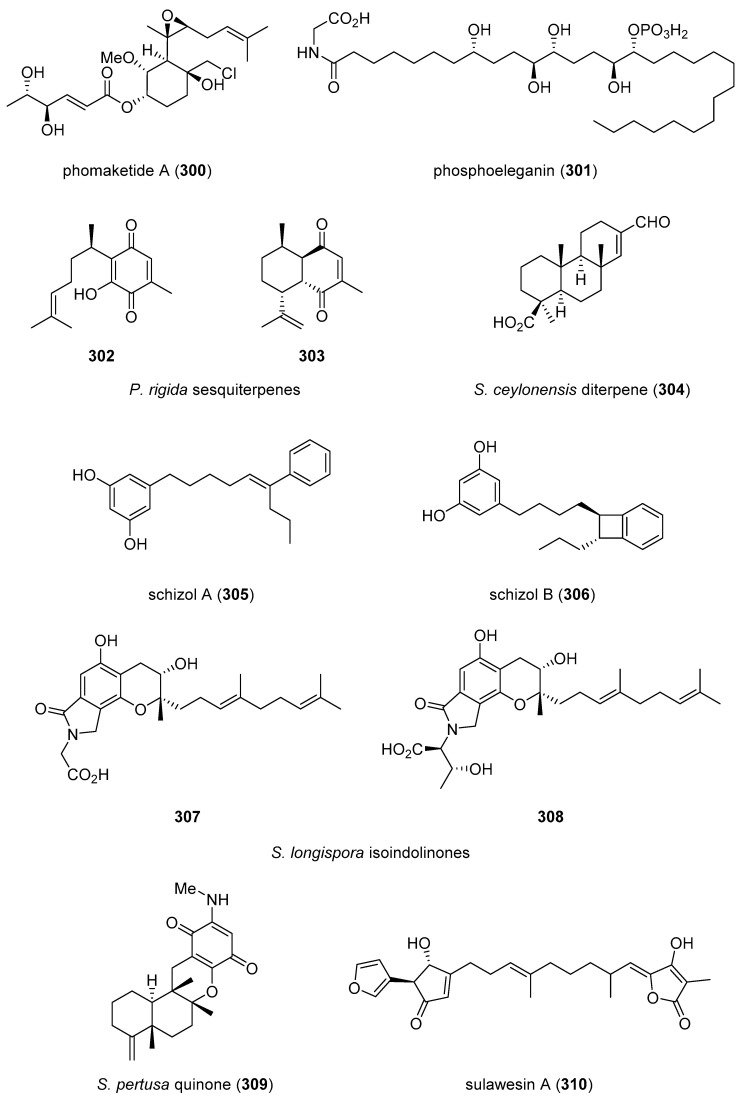

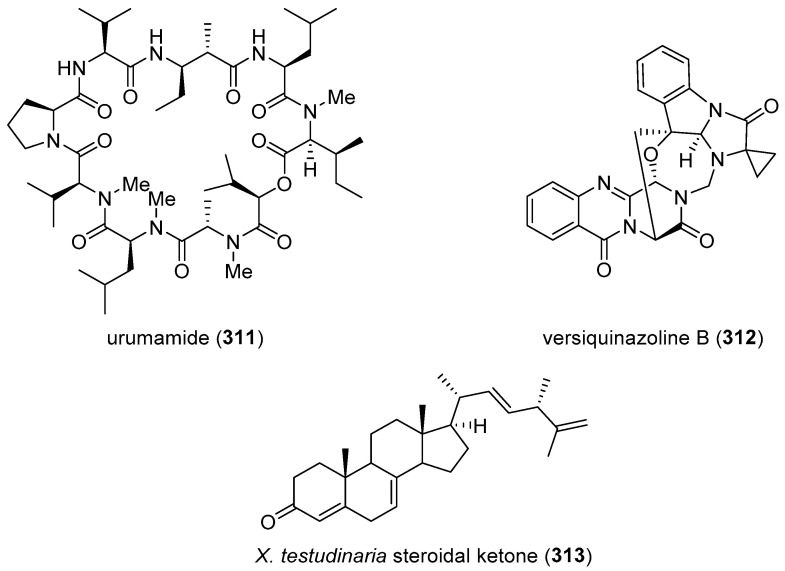



## 5. Reviews on Marine Pharmacology and Pharmaceuticals

In 2016–2017 several reviews covered general and/or specific areas of marine preclinical pharmacology: (a) *marine pharmacology and marine pharmaceuticals:* new marine natural products and relevant biological activities published in 2016 and 2017 [330,331]; chemistry and biology of guanidine natural products [332,333]; biological properties of secondary metabolites from sea hares of *Aplysia* genus [334]; alkynyl-containing peptides of marine cyanobacteria and molluscs [335]; bioactive cyanobacterial secondary metabolites for health [336]; biological active metabolites from marine-derived myxobacteria [337]; antimicrobial metabolites from the marine bacteria genus *Pseudoalteromonas* [338]; marine natural products from marine-derived *Penicillium* fungi [339]; biological activity of secondary metabolites from marine-algal-derived endophytic fungi [340]; pharmacological potential of fucosterol from marine algae [341]; pharmacological activities of Antarctic marine natural products [342]; bioactive acetylated triterpene glycosides from Holothuroidea in the past six decades [343]; terpenoids from octocorals of the genus *Pachyclavularia* [344]; bioactive marine natural products from sponges of the genus *Hyrtios* [345]; secondary metabolites from the marine sponge genus *Phyllospongia* [346]; discovery strategies of bioactive compounds synthetized by nonribosomal peptide synthetases and type-1 polyketide synthase derived from marine microbiomes [347]; developing natural product drugs: supply problems and how they have been overcome [348]; the global marine pharmaceutical pipeline in 2020: U.S. Food and Drug Administration-approved compounds and those in Phase I, II and III of clinical development http://marinepharmacology.midwestern.edu/clinPipeline.htm; (b) *antimicrobial marine pharmacology:* antimycobacterial metabolites from marine invertebrates [349]; antimicrobials from cnidarians [350]; (c) *antiprotozoal and antimalarial marine pharmacology:* natural products in drug discovery against neglected tropical diseases [351]; antimycobacterial natural products from marine *Pseudopterogorgia elisabethae* [352]; (d) *immuno- and anti-inflammatory marine pharmacology:* marine natural products inhibitors of neutrophil-associated inflammation [353]; (e) *cardiovascular and antidiabetic marine pharmacology*: bioactive components from fish for dyslipidemia and cardiovascular risk reduction [354]; (f) *nervous system marine pharmacology*: marine natural products from sponges with neuroprotective activity [355]; a transcriptomic survey of ion channel-based conotoxin in the Chinese cone snail *Conus betulinus* [356]; dinoflagellate cyclic imine toxins as potent antagonists of nicotinic acetylcholine receptors [357]; inhibition of nociception and pain transmission by analgesic conopeptides ion channel inhibition by targeting G protein-coupled receptors [358]; (g) *miscellaneous molecular targets and uses*: ichthyotoxicity evaluation of marine natural products [359]; pharmacological potential of non-ribosomal peptides from marine sponges and tunicates [360]; aeroplysinin-1 as a multi-targeted bioactive sponge-derived natural product [361]; therapeutic potential of the phycotoxin yessotoxin [362], and new modalities for challenging drug targets in pharmaceutical discovery [363].

## 6. Conclusions

This marine pharmacology 2016–2017 review is the eleventh contribution to the marine preclinical pharmacology pipeline review series that was initiated by Alejandro M. S. Mayer (AMSM) in 1998 [1,2,3,4,5,6,7,8,9,10], with the aim of providing a systematic overview of selected peer-reviewed preclinical marine pharmacological literature. The global preclinical marine pharmacology research highlighted in this review involved chemists and pharmacologists from 54 countries, a remarkable increase from our last review, namely: Australia, Austria, Bangladesh, Belgium, Brazil, Canada, China, Costa Rica, Croatia, Cuba, Denmark, Egypt, Fiji, France, Germany, Greece, Hungary, Iceland, India, Indonesia, Ireland, Israel, Italy, Japan, Jordan, Kazakhstan, Malaysia, Mexico, Morocco, Myanmar, the Netherlands, Nigeria, New Zealand, Norway, Oman, Panama, Papua New Guinea, Philippines, Poland, Portugal, Russian Federation, Saudi Arabia, Serbia, Slovenia, South Korea, Spain, Sweden, Switzerland, Taiwan, Thailand, Turkey, United Kingdom, United States and Vietnam. Thus, during 2016–2017 the marine *preclinical* pharmaceutical pipeline continued to provide novel pharmacology and potential new leads for the marine *clinical* pharmaceutical pipeline. As shown at the marine *preclinical* and *clinical* pharmaceutical pipeline website: https://www.midwestern.edu/departments/marinepharmacology.xml there are currently 13 marine-derived pharmaceuticals approved by the U.S. Food and Drug Administration and 1 by Australia, and more than 23 marine-derived compounds in Phases I, II and III of global *clinical* pharmaceutical development.

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
