# Peer review of "Marine Pharmacology in 2016–2017: Marine Compounds with Antibacterial, Antidiabetic, Antifungal, Anti-Inflammatory, Antiprotozoal, Antituberculosis and Antiviral Activities; Affecting the Immune and Nervous Systems, and Other Miscellaneous Mechanisms of Action"

_marinedrugs, 2021, doi:10.3390/md19020049_

Round 1
Reviewer 1 Report
This is a continuation of a very useful series of reviews of the field of marine natural products written by a seasoned group of authors that will be useful to all investigators. The manuscript has been carefully organized and is remarkably free of grammarical problems!
My only suggestion is that some paragraphs be split into two or more paragraphs in an effort to further organize the literature and findings, and that some topics not be lumped together.
One typo occurs on page 2, line 60: "for" should be "from".
Author Response
Manuscript ID Marinedrugs-1050267
Mayer, A.M.S., A.J. Guerrero, A. D. Rodríguez, O. Taglialatela-Scafati, F. Nakamura and N. Fusetani. Marine Pharmacology in 2016–2017: Marine Compounds with Antibacterial, Antidiabetic, Antifungal, Anti-Inflammatory, Antiprotozoal, Antituberculosis, and Antiviral Activities; Affecting the Immune and Nervous Systems, and other Miscellaneous Mechanisms of Action.
Response to Reviewer 1:
“This is a continuation of a very useful series of reviews of the field of marine natural products written by a seasoned group of authors that will be useful to all investigators. The manuscript has been carefully organized and is remarkably free of grammatical problems! My only suggestion is that some paragraphs be split into two or more paragraphs in an effort to further organize the literature and findings, and that some topics not be lumped together.
One typo occurs on page 2, line 60: "for" should be "from".
We thank the reviewer for carefully reading our manuscript and for his/her very kind words. This review is our eleventh marine pharmacology review of the non-cancer marine natural products which attempts to present a systematic overview of the pharmacological activity of novel marine natural products other than antitumor and cytotoxic activities. Based on feedback we have received from many readers, we believe that our work contributes a distinct approach to the systematic review of the expanding literature on the pharmacology of marine natural products. In contrast, as the reviewer is well aware, the chemistry of marine natural products has been reviewed systematically by Blunt et al. Our objective is to provide a comprehensive review of peer-reviewed articles published in 2016-2017, and which have been carefully selected from a total of 940 papers in our 2016-2017 Endnote database, thus emphasizing the more promising and significant pharmacological results. We are hopeful the reviewer will be satisfied with the minor changes introduced to the revised manuscript.
Specific comments
- One typo occurs on page 2, line 60: "for" should be "from".
We thank the reviewer for his/her comments and suggestions for improvement of our review manuscript. We have proceeded to review line 60 in the manuscript. These specific changes in the revised manuscript are highlighted in yellow.
- The following text in line 60 has been revised:
activity” [15]; antibacterial activity in extracts from the marine bacterium Salinispora arenicola for the Gulf of California, Mexico [16];
The revised text in line 60, now reads:
activity” [15]; antibacterial activity in extracts from the marine bacterium Salinispora arenicola from the Gulf of California, Mexico [16];
We hope the reviewer is now satisfied with all the revisions we have made to the manuscript, and is willing to accept the revised manuscript. We again wish to apologize for this mistake in the submitted manuscript which we have carefully corrected, and finally were are very grateful for the time the reviewer has spent in helping us improve our 2016-2017 marine pharmacology review manuscript.
Reviewer 2 Report
This review is an exhaustive examination of bioactivity in marine compounds reported from 2016-2017. The addition of mechanisms of action, where known, was appreciated. The only typo I noted as on line 475; "detectd" should be "detected." Other than that, this manuscript should be published as is.
Author Response
Manuscript ID Marinedrugs-10502677
Mayer, A.M.S., A.J. Guerrero, A. D. Rodríguez, O. Taglialatela-Scafati, F. Nakamura and N. Fusetani. Marine Pharmacology in 2016–2017: Marine Compounds with Antibacterial, Antidiabetic, Antifungal, Anti-Inflammatory, Antiprotozoal, Antituberculosis, and Antiviral Activities; Affecting the Immune and Nervous Systems, and other Miscellaneous Mechanisms of Action.
Response to Reviewer 2:
“This review is an exhaustive examination of bioactivity in marine compounds reported from 2016-2017. The addition of mechanisms of action, where known, was appreciated. The only typo I noted was on line 475; "detectd" should be "detected." Other than that, this manuscript should be published as is.”
We thank the reviewer for carefully reading our manuscript and for his/her very kind words. This review is our eleventh marine pharmacology review of the non-cancer marine natural products which attempts to present a systematic overview of the pharmacological activity of novel marine natural products other than antitumor and cytotoxic activities. Based on feedback we have received from many readers, we believe that our work contributes a distinct approach to the systematic review of the expanding literature on the pharmacology of marine natural products. In contrast, as the reviewer is well aware, the chemistry of marine natural products has been reviewed systematically by Blunt et al. Our objective is to provide a comprehensive review of peer-reviewed articles published in 2016-2017, and which have been carefully selected from a total of 940 papers in our 2016-2017 Endnote database, thus emphasizing the more promising and significant pharmacological results. We are hopeful the reviewer will be satisfied with the minor changes introduced to the revised manuscript.
Specific comments
- The only typo I noted was on line 475; "detectd" should be "detected."
We thank the reviewer for his/her comments and suggestions for improvement of our review manuscript. We think the reviewer was referring to the word “decreasd” rather than “detect“ on line 475. We have therefore proceeded to review line 475 in the manuscript. These specific changes in the revised manuscript are highlighted in yellow.
- The following text in line 475 has been revised:
“…and in mice “significantly decreasd the blood glucose”, thus suggesting BDDE might be a “ treatment…”
The revised text in line 475, now reads:
…”and in mice “significantly decreased the blood glucose”, thus suggesting BDDE might be a “ treatment…”
We hope the reviewer is now satisfied with the revision we have made to the manuscript, and is willing to accept the revised manuscript. We again wish to apologize for our mistake in the submitted manuscript which we have now been carefully corrected. Finally were are very grateful for the time the reviewer has spent carefully reviewing our marine pharmacology review manuscript.